# Focus Directions Make Your Language Models Pay More Attention to Relevant Contexts

## Abstract

Long-context large language models (LLMs) are prone to being distracted by irrelevant contexts. The reason for distraction remains poorly understood. In this paper, we first identify the contextual heads, a special group of attention heads that control the overall attention of the LLM to the contexts. Then, we demonstrate that distraction arises when contextual heads fail to allocate sufficient attention to relevant contexts and can be mitigated by increasing attention to these contexts. We further identify focus directions, located at the key and query activations of these heads, which control the amount of attention activated from the attention sink to the contexts. With a proper amount of attention activation, the contextual heads could allocate more attention to relevant contexts. Motivated by this, we introduce an automated magnitude control method that keeps attention activation within a proper range, enabling practical use of focus directions. We comprehensively evaluate the effect of focus direction on various long-context tasks and find that focus directions can help mitigate the poor task alignment of long-context LLMs. We believe our findings could promote further research on long-context LLM alignment.

## 1 Introduction

Long-context large language models enable multiple applications, such as many-shot in-context learning Li et al. (2024c); Agarwal et al. (2025); Bertsch et al. (2024), summarization Chang et al. (2023); Kim et al. (2024), and retrieval-augmented generation Lee et al. (2024). Given a long context window such as 128k tokens, only a small amount of the contexts are relevant to the task, and a large amount of contexts are irrelevant. Long context LLM may be distracted by irrelevant contexts Liu et al. (2024); Shi et al. (2023). Such distractions may result in generating false information, erroneous reasoning, and negative social impacts.

The reason for LLMs being distracted by irrelevant context is poorly understood. In this paper, we aim to reveal the **cause of the distraction** (§2). As shown in Figure 1, starting with a dataset with labels of relevant and irrelevant context, we first introduce a contextual scoring method, which measures the strength of the attention to the relevant context during text generation. Based on such a scoring method, we identify **contextual heads**, a special group of attention heads with the highest score. We then adjust the strength of attention of these heads to the relevant contexts based on the label of the relevant context. We found that increasing attention on these heads to the relevant context increases the downstream task performance, while decreasing attention decreases the performance. While other non-contextual heads have minimal such effects. We conclude that contextual heads could control the overall attention of LLMs to the contexts.

Building upon the findings of the contextual head, we further identify the **focus directions** (§3), which control how much attention is being activated from attention sinks Xiao et al. (2023) to the contexts. Focus directions are located at the key and query activations of the contextual heads. Similar to other directional vectors, like refusal Arditi et al. (2024), sentiment Han et al. (2023), and truthfulness Li et al. (2024b), we found that applying a proper magnitude of focus directions could enable LLMs to pay more attention to the relevant contexts, and thus improve the downstream performance.

To understand how focus directions affect the capability of long-context LLMs (§4), we apply focus directions to three families of LLMs and evaluate them on HELMET Yen et al. (2024), a comprehensive long-context task benchmark. We found that focus directions could help mitigate poor task alignment of the LLMs. At last, we discuss the potential application of the focus directions.

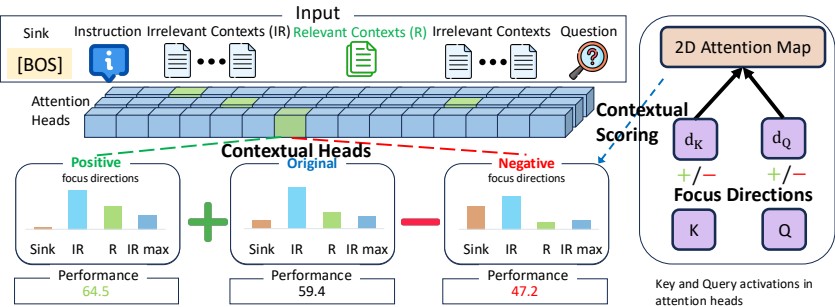

Figure 1: Overview of this work. We first introduce **contextual scoring**, measuring the attention distribution over inputs during response generation. Based on contextual scoring, we identify the **contextual heads**, which control the overall attention of LLMs. We further find out **focus directions**, which make LLMs pay more attention to the relevant contexts.

## 2 CAUSE OF DISTRACTION

To reveal the cause of LLMs being distracted by irrelevant contexts, we first identify the attention heads that are mostly responsible for extracting information from relevant contexts, which we named contextual heads §2.1. Then, we study the basic properties of the contextual heads, including their location and behavior in different cases §2.2. At last, we demonstrate that increasing attention to relevant contexts on these heads could mitigate distractions §2.3.

### 2.1 IDENTIFYING CONTEXTUAL HEADS

To identify contextual heads, we introduce a contextual scoring method to identify the attention distribution of different parts of input for each attention head in the transformer architecture. Our method is based on the Multi-Document Question Answering (QA) data introduced by the "lost in the middle" paper Liu et al. (2024).

**Multi-Document Question Answering data.** The data is initiated with the NaturalQuestions-Open data Lee et al. (2019); Kwiatkowski et al. (2019). Each samples have a question and a list of answers. The questions are user queries from Google search, and the answers are human-annotated based on Wikipedia. The authors of Liu et al. (2024) further matched each question and answer pair with a set of documents using a retrieval system. In these documents, only one contains the answer (i.e., relevant context), and others do not contain the answer (i.e., irrelevant contexts).

**Experiment settings.** The above dataset has 2654 samples in total, we randomly split them into half training and half testing. The input is defined as $P = [I_p, \hat{C}_{before}, C, \hat{C}_{after}, I_q]$, where $I_p$ and $I_q$ are instructions, specifying the QA task (e.g., a system prompt and a question). The $C$ stands for relevant context, $\hat{C}_{before}$, and $\hat{C}_{after}$ stands for the irrelevant contexts before and after the relevant context, which can be zero, one, or more documents. We consider 20 document cases where one of the documents in the input is relevant and the rest of the 19 documents are irrelevant. We put the relevant documents in positions 1, 5, 10, 15, and 20. The input is fed into an LLM, in our case, we use Llama-3.2-3B instruction model[1], to obtain an LLM response $R$ using greedy decoding. The evaluation metric is the exact match (EM) accuracy. If the model output matches one of the answers in the output list, then it is considered to be correct; otherwise, it is wrong.

**Contextual scoring.** Based on the above data and experiment settings, we introduce the following contextual scoring method, which aims to find a set of attention heads in the LLM that pay the most attention to the relevant contexts during generation. Let $W \in \mathbb{R}^{T \times T}$ be the attention weight matrix of an attention head, where $T$ is the sequence length. For each token $r_i$ in the generated response $R = [r_{start}, \ldots, r_{end}]$, we extract the attention weights corresponding to relevant contexts

---

[1] https://huggingface.co/meta-llama/Llama-3.2-3B-Instruct

| | Long | | | | Correct | | | | Wrong | | | | Gold | |
|---|---|---|---|---|---|---|---|---|---|---|---|---|---|---|
| Heads | R↑ | IR↓ | IR max↓ | Sink | R↑ | IR↓ | IR max↓ | Sink | R↑ | IR↓ | IR max↓ | Sink | R | Sink |
| (13, 23) | 0.209 | 0.516 | 0.160 | 0.105 | 0.290 | 0.437 | 0.125 | 0.114 | 0.106 | 0.612 | 0.202 | 0.093 | 0.555 | 0.187 |
| (12, 1) | 0.203 | 0.568 | 0.161 | 0.079 | 0.283 | 0.490 | 0.129 | 0.084 | 0.106 | 0.664 | 0.201 | 0.070 | 0.637 | 0.153 |
| (15, 18) | 0.199 | 0.423 | 0.138 | 0.254 | 0.279 | 0.338 | 0.101 | 0.267 | 0.101 | 0.525 | 0.183 | 0.238 | 0.507 | 0.317 |
| (15, 22) | 0.195 | 0.391 | 0.140 | 0.244 | 0.277 | 0.309 | 0.103 | 0.254 | 0.098 | 0.487 | 0.184 | 0.227 | 0.481 | 0.280 |
| (14, 2) | 0.185 | 0.339 | 0.130 | 0.294 | 0.270 | 0.262 | 0.086 | 0.278 | 0.080 | 0.429 | 0.181 | 0.311 | 0.345 | 0.458 |

Table 1: Contextual scores of top-5 contextual heads. **Heads**: (Layer, head number), **R**: relevant contextual score, **IR**: irrelevant contextual score, **IR max**: max single document irrelevant contextual score, **Sink**: sink contextual score. We consider four cases: **Long**: standard 20-documents long context case. **Gold**: with only relevant contexts but not irrelevant ones. **Correct**: exactly matched for both gold and long case. **Wrong**: exactly matched for the gold case but not exactly matched in the long case. We define the correct and wrong based on the gold to filter out the cases that are not doable for LLMs.

$C = [c_{start}, \ldots, c_{end}]$ and sum over this span, and then average through each response token $r_i$:

$$S_C = \frac{1}{|R|} \sum_{i=r_{start}}^{r_{end}} \sum_{j=c_{start}}^{c_{end}} W_{i,j} \tag{1}$$

This score quantifies how much an attention head focuses on the relevant context while generating the response. Higher values indicate stronger attention toward the relevant span, helping to identify heads that extract the most information from the relevant contexts. We then further average the score $S_C$ through the dataset for each head, obtaining a **relevant contextual score**. We do not normalize the score by length since, at the dataset level, each document does not have a significant difference in length. With such a score, we are now able to identify the **contextual heads** with top-$k$ scores focused on the relevant contexts. Similarly, we can extend the definition of the relevant contextual score to any text span in the input. We could define **irrelevant contextual score**, which measures the attention to the entire irrelevant contexts (i.e., $\hat{C}_{before}$ and $\hat{C}_{after}$); **max single document irrelevant contextual score**, which represents the highest contextual score among individual documents within the irrelevant contexts; **sink contextual score**, which measure the "dummy" attention to the attention sink (i.e., starts tokens) Xiao et al. (2023) when that part of attention do not need to pay in other non-start tokens.

## 2.2 PROPERTIES OF CONTEXTUAL HEADS

**Contextual heads are sparse.** As shown in Figure 2, among 672 attention heads in Llama-3.2-3B instruction model, only 2 (0.3%) of the heads have a relevant contextual score that > 0.2. Also, only 37 (5.5%) of the heads have a relevant contextual score >0.1, and only 113 (16.8%) of the heads have a relevant contextual score >0.05. In general, only a small amount of heads with high relevant contextual scores are considered to extract information from relevant contexts during autoregressive generation. Most heads, with low relevant contextual scores, are not considered to extract information from the relevant contexts.

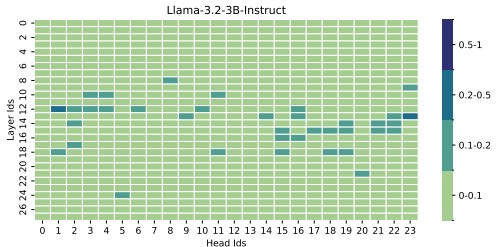

Figure 2: Location of the contextual heads.

**Contextual heads are mostly located in middle and late layers.** As shown in Figure 2, most of the contextual heads with relevant contextual scores >0.1 are located from layer 8 to layer 18 (index from 0 to 27).

**Contextual heads focus more on relevant context when the response is correct, focus less on relevant context when the response is wrong.** As shown in Table 1, we found that overall, relevant contexts have lower scores than the irrelevant contexts since we have 19 documents as irrelevant context and only 1 as relevant context. However, in the long and correct case, the score for relevant context is larger than the IR max score. This means contextual heads have more focus on the relevant

context when the generated answer is correct. While in the wrong case, this does not hold that relevant contexts have a lower score than the irrelevant ones with a max score.

**More attention is "activated" for long contexts compared to the short ones.** As shown in Table 1, sink contextual scores are similar for long, correct, and wrong cases. However, the gold has a higher sink contextual score than the other three long context cases. At the same time, less attention is paid to the contexts for the gold cases than the three long context cases since the attentions are summed up to 1. This suggests that more attention is "activated" for long contexts compared to the short ones, and the sink contextual scores could be an indicator for such activation.

## 2.3 ATTENTION COMPENSATION ON CONTEXTUAL HEADS

From §2.2 we demonstrate the correct cases have a higher attention to the relevant contexts compared to the wrong cases. In this section, we aim to further demonstrate that if we could increase the attention to the relevant contexts for the contextual head, the distraction could be mitigated.

**Attention compensation method.** We use split-softmax Li et al. (2024a), which can increase or decrease the attention on a token span for some specific attention heads. Specifically, given the attention weight matrix $W \in \mathbb{R}^{T \times T}$ at layer $\ell$ and head $h$, we aim to modify the attention weights assigned to the relevant context span $C = [c_{start}, \ldots, c_{end}]$. First, for each response token $r_i$ to be generated, we compute the total attention allocated to the span $C$ by summing the relevant attention weights:

$$\pi_C(i) = \sum_{j=c_{start}}^{c_{end}} W_{i,j} \quad (2)$$

Figure 3: Performance across different top-$k$ contextual/random heads and split softmax exponents $\tau$. Baseline: 20 documents (1 relevant, 19 irrelevant) case without intervention. Gold baseline: 1 relevant document case without intervention. Negative baseline: 19 irrelevant documents case without intervention.

We then rescale the attention distribution using the split-softmax transformation:

$$W'_{i,j} = \begin{cases} \frac{\pi_C(i)^\tau}{\pi_C(i)} W_{i,j}, & \text{if } j \in C \\ \frac{1-\pi_C(i)^\tau}{1-\pi_C(i)} W_{i,j}, & \text{if } j \notin C \end{cases} \quad (3)$$

where $\tau$ is the split softmax exponent controlling the strength of the modification, with $\tau \geq 0$. When $0 \leq \tau < 1$, attention is increased for the span $C$, when $\tau = 1$, no modification is applied, and when $\tau > 1$, attention is decreased for the span $C$. And smaller values of $\tau$ increase the attention, while larger values of $\tau$ decrease the attention. The reweighted matrix $W'$ ensures that the attention scores still sum to 1 across each row while redistributing more attention toward the span $C$.

**Experiment settings.** We experiment with split softmax exponent $\tau = (0.1, 0.3, 0.6, 1.5, 1000)$ with the top-$k$ heads of $(1, 5, 10, 20, 30, 50, 100, 150, 200, 300, 400, 500, 600)$, using the testing split of our dataset. We also report the baseline EM accuracy of 0.59, which is without any split softmax intervention.

**Increasing attention to the relevant contexts mitigates the distraction, while decreasing attention to the relevant contexts results in more distraction.** As shown in Figure 3, increasing the attention to the relevant contexts ($\tau < 1$) improves the performance. For all the cases of $\tau = (0.1, 0.3, 0.6)$, the EM accuracy is larger than the baseline. On the other hand, decreasing the attention to the relevant contexts ($\tau > 1$) decreases the performance.

**Increasing attention on the contextual heads mitigates distraction, while increasing attention on non-contextual heads has a limited effect on distraction mitigation.** We demonstrate this through

two aspects: using top-$k$ contextual heads and $k$ random heads. As shown in Figure 3, for the top-$k$ contextual heads, for all cases of $\tau < 1$, the EM accuracy improves with more attention heads being intervened from top-1 to top-20. The best EM accuracy (0.916) is achieved with top-20 heads and $\tau = 0.1$. However, with more top-$k$ heads intervened, the EM accuracy is decreased compared to the top-20 case. Notably, adding too much attention ($\tau = 0.1$) on 600 heads even makes the EM accuracy drop under the baseline. On the other hand, when using $k$ random heads with $\tau = 0.3$, we observe a limited ($<0.3\%$) EM accuracy improvement with $<20$ heads, a performance drop when using 50 heads, and a similar performance compared to contextual heads when using more than 400 heads. This demonstrates that increasing attention helps more with distraction mitigation when using contextual heads and helps less when using non-contextual heads.

**Contextual heads control the overall attention of the LLM.** As shown in Figure 3, when intervening in top-20 contextual heads, increasing attention to the relevant context on the contextual heads, the EM accuracy can reach up to 0.916, better than the gold baseline of 0.847. On the other hand, with decreasing attention to the relevant context on the contextual heads, the EM accuracy can drop to 0.320, close to the negative baseline of 0.276. This suggests that the contextual heads control the overall attention of the LLM to the input tokens. In the case of increased attention on the contextual heads, the effect of input tokens in the relevant contexts can be amplified. In case of decreased attention on the contextual heads, the effect of input tokens in the relevant contexts can be nullified.

## 3 ELICITING ATTENTION ON RELEVANT CONTEXTS VIA FOCUS DIRECTION

From §2.3 we show that increasing attention on the relevant contexts could mitigate the distraction. However, in practice, we do not have the label of relevant contexts during LLM inference. We wonder, can contextual heads figure out the relevant contexts by themselves? Inspired by previous direction addition works Turner et al. (2023); Arditi et al. (2024); Li et al. (2024b), we hypothesize the existence of a focus direction that could make LLMs focus more on the relevant contexts. In this section, we first introduce a method to obtain the focus directions (§3.1). Then, we discuss the usage and effect of the focus directions (§3.2).

### 3.1 OBTAINING FOCUS DIRECTION

To obtain the focus direction, we first need to identify the location of the focus direction. Previous works mainly focused on the residual stream activation Turner et al. (2023); Arditi et al. (2024) or O projection Li et al. (2024b) of attention heads, which do not have a direct relation with the attention and may not be feasible for our case. Since the attention is produced by key and query activation, we hypothesize that focus directions are situated within the key and query representation spaces. Based on the hypothesis, we aim to find two focus direction vectors, one for key activation and another for query activation for each attention head.

**Obtain focus directions by training.** We consider a simple training method to obtain the focus direction. We first generate a response with $[I_p, C, I_q]$ (i.e., with relevant context only), obtain a gold LLM response $R_g$, for each sample in our training split, and obtain text sequences $[I_p, C, I_q, R_g]$. We then cache the key activations $K \in \mathbb{R}^{T \times F}$ and query activations $Q \in \mathbb{R}^{T \times F}$ of the text sequence for each attention head, where $F$ is the feature dimension of $Q$ and $K$. The original attention weights is obtained by $W = \mathrm{softmax}\left(\frac{QK^\top}{\sqrt{F}}\right)$. We add focus direction vectors $d_K \in \mathbb{R}^F$ and $d_Q \in \mathbb{R}^F$ for $K$ and $Q$, obtaining a new attention weights

$$W^d = \mathrm{softmax}\left(\frac{(Q + d_Q)(K + d_K)^\top}{\sqrt{F}}\right) \tag{4}$$

Given the new $W_d$, we can simply put it into Equation 1, which obtains $S_C^d = \frac{1}{|R|} \sum_{i=r_{start}}^{r_{end}} \sum_{j=c_{start}}^{c_{end}} W_{i,j}^d$, measuring the attention to the relevant contexts $C$ when generating the LLM answer. We can use a simple loss function $L = -S_C^d$, training $d_K$ and $d_Q$ to obtain the focus direction. The directions maximize attention to the relevant contexts of the corresponding attention head during the response generation process.

### 3.2 INFERENCE TIME INTERVENTION WITH FOCUS DIRECTION

Given focus direction $d_K$ and $d_Q$ for an attention head obtained by the previous step, we can apply them at inference time with the following:

$$W = \text{softmax}\left(\frac{(Q + \alpha d_Q)(K + \alpha d_K)^\top}{\sqrt{F}}\right) \quad (5)$$

where $\alpha$ is an intervention magnitude factor to control the magnitude of the intervention. When $\alpha > 0$ is the positive intervention, aim to make the attention head pay more attention to the relevant context. When $\alpha < 0$ is the negative intervention, aim to make the attention head pay less attention to the relevant context. When $\alpha = 0$ no intervention is applied. In addition, we can have a hyperparameter $k$ that intervenes top-$k$ contextual heads.

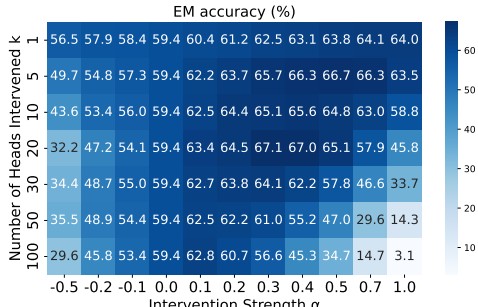

Figure 4: EM accuracy of different top-$k$ heads and $\alpha$.

### 3.3 EXPERIMENT SETTINGS

We first cache the activations for the whole sequence of our training split and then obtain the focus directions by training. We used AdamW optimizer with a learning rate of $10^{-3}$ training for 10 epochs. For evaluation, we used our testing split. We report the contextual scores of the top-5 heads in Table 3 and the EM accuracy in Figure 4.

### 3.4 RESULTS

**Focus directions make contextual heads pay more attention to the relevant context.** As shown in Table 3, when a positive focus direction is applied ($\alpha = 0.2$ and $\alpha = 0.5$), the contextual scores on the relevant context are increased. Also, the higher the $\alpha$, the more attention to the relevant contexts. On the other hand, when a negative focus direction is applied, the contextual scores on the relevant context are decreased.

**Focus directions control attention activation from the sink.** As shown in Table 3, while increasing the attention to the relevant contexts, positive focus directions do not decrease the attention to the irrelevant contexts. Instead, the attention on irrelevant context may still have little increase. The main attention reassigned to the relevant contexts is from the attention sink. This suggests the main function of positive focus direction is to move the attention from the sink to the relevant contexts. On the other hand, if a negative focus direction is applied, the amount of attention in the attention sink is increased.

**Positive focus direction mitigates distraction, while negative focus direction leads to more distraction.** As shown in Figure 4, when applying a positive focus direction with $0 < \alpha \leq 0.5$, for the top 1-20 heads, the EM accuracy has a consistent improvement compared to the baseline (59.4 %). The best EM accuracy of 67.1% was achieved with $\alpha = 0.3$ with top-20 heads[2]. This demonstrates that positive focus directions could mitigate distraction. On the other hand, when applying a negative focus direction with $\alpha < 0$, the EM accuracy drops under the baseline, indicating more distraction than no intervention.

**Focus directions only help mitigate distraction on contextual heads.** When applying a positive focus direction, we observe that an intervention of $> 20$ heads always results in lower EM accuracy than the one of 20 heads. This indicates focus direction only helps mitigate distraction on contextual heads. Applying focus direction on non-contextual heads may not help mitigate distraction. The observation is also consistent with the attention compensation result in Figure 3.

**Applying overly strong focus directions can inadvertently heighten attention to irrelevant contexts.** As shown in Table 3, from $\alpha = 0.4$ to $\alpha = 0.5$, the IR max score starts to rise at a higher rate than the R score. For example, for the head (13, 23), the R score increased from 0.40 to 0.41, and the IR max score increased from 0.21 to 0.23. The raised IR max score distracts the LLM, making

---

[2]As noted in Liu et al. (2024), some distractor passages may contain a reasonable answer. As such, we don't expect the EM accuracy here to be comparable with the one in Figure 3.

the corresponding EM accuracy drop from 67.0% to 65.1%. Furthermore, when $\alpha = 1.0$, the R score further drops to 0.34, and its value is similar to the IR max score. And the corresponding EM accuracy dropped 45.8%, even worse than the baseline of 59.4%. This indicates that applying a strong focus direction can also distract the LLM. The amount of attention activation is needed to align LLMs to achieve optimal downstream performance.

### 3.5 A PROPER AMOUNT OF ATTENTION ACTIVATION MATTERS

In the previous section, we discussed the results of applying focus direction intervention from a dataset-level view. Here, we further discuss this from a single-sample perspective.

**Improper amount of attention activation breaks the attention distribution.** An improper amount of attention activation could be either too large or too small. An overly strong positive focus direction could result in an attention activation level that is too high, and an overly strong negative focus direction could lead to an activation level that is too low. Specifically, the focus direction can be applied as follows: $(Q + \alpha d_Q)(K + \alpha d_K)^\top$ (Numerator part of Equation 5). And it can be expanded to $QK^\top + \alpha Q d_K^\top + \alpha d_Q K^\top + \alpha^2 d_Q d_K^\top$. Where the $QK^\top$ is the original attention weights before normalization, the rest are the extra terms related to the focus directions. In case of overly strong focus direction (either positive or negative), the extra terms may be larger than the $QK^\top$, which completely breaks the attention distribution of the LLMs. From the dataset-level view, the larger the $\alpha$, the more samples will have the attention distribution broken, and thus the performance will decrease. For example, in Figure 4, the 20-head case, the performance starts to decrease when $\alpha > 0.3$.

**Automated magnitude control.** To control the amount of attention activation in a proper range, we provide an initial exploration of automated magnitude control using a magnitude controller. We consider two types of training objectives for automated magnitude control: an aggressive Median Intervention (MI) objective and a conservative Intervention when Necessary (IN) objective. The details are in §C. We found that automated magnitude control improves downstream performance without requiring manual specification of the magnitude of intervention. Such a method enables focus directions applicable in real-world applications.

## 4 FOCUS DIRECTIONS ARE GENERALIZABLE TO DIFFERENT TASKS

To study the effect of the focus direction on various long-context tasks, we use HELMET Yen et al. (2024), a comprehensive benchmark for long-context evaluation. We use five categories of the task from HELMET, including **Synthetic recall (Recall)** (needle-in-a-haystack Hsieh et al. (2024) and JSON KV retrieval task Liu et al. (2024)), **Retrieval-augmented generation (RAG)** (KILT benchmark Petroni et al. (2020), including Natural Questions (NQ) Kwiatkowski et al. (2019), TriviaQA Joshi et al. (2017), HotpotQA Yang et al. (2018), PopQA Mallen et al. (2022)), **Passage re-ranking (Re-rank)** (MS MARCO Bajaj et al. (2016)), **Many-shot in-context learning (ICL)** (TREC-course, TREC-fine Li & Roth (2002), BANKING77 Casanueva et al. (2020), CLINC150 Larson et al. (2019), NLU Liu et al. (2021)), **Long-document QA (Long QA)**(Infbench QA and multiple choice (MC) Zhang et al. (2024)).

**Experiment settings.** We consider three LLMs, including Llama-3.2-3B-Instruct, Qwen2.5-7B-Instruct, and Ministral-8B-Instruct-2410. To show the effect of focus direction on base models, we also provide the results of Llama-3.2-3B and Qwen2.5-7B, using the focus direction obtained by their corresponding instruction models. We consider five settings, including baseline (no intervention), $\alpha = -0.2$, and 0.2 for top-10 and top-20 attention heads. Also, we experiment with 8k, 16k, 32k, 64k, and 128k token contexts, following the HELMET benchmark. We report the 32k and 64k results in Table 2, and the rest are in the tables in the appendix. We also report the sink contextual score under 8k and 16k contexts in Table 13 and 14.

**Focus direction mitigates poor task alignment.** We discuss this from two aspects. First, we compare the task performance between base models and instruction models. For a task, if there is a performance gain after post-training, the base model may have a performance gain by applying the focus direction. For example, as shown in Table 5, for the HotpotQA task under 8k contexts (Llama), the performance improved from 52.67% (base model) to 62.00% after post-training. When focus directions are applied, the base model performance could be improved to 56.00%. In this case,

| Model | Recall | RAG | Re-ranking | ICL | Long QA | Overall Average | Model | Recall | RAG | Re-ranking | ICL | Long QA | Overall Average |
|---|---|---|---|---|---|---|---|---|---|---|---|---|---|
| **Llama-3.2-3B** | | | | | | | **Llama-3.2-3B** | | | | | | |
| 20_-0.2 | 82.19 | 56.25 | 30.15 | 75.20 | - | 60.95 | 20_-0.2 | 66.00 | 54.96 | 29.22 | 82.20 | - | 58.10 |
| 10_-0.2 | 88.19 | 58.67 | 32.46 | 75.20 | - | 63.63 | 10_-0.2 | 73.81 | 56.58 | 26.73 | 83.00 | - | 60.03 |
| 20_0.2 | 87.81 | 60.29 | 31.44 | 76.60 | - | 64.04 | 20_0.2 | 81.50 | 58.75 | 25.37 | 80.20 | - | 61.46 |
| 10_0.2 | 89.81 | 60.38 | 31.54 | 77.40 | - | 64.78 | 10_0.2 | 82.00 | 58.54 | 26.16 | 80.60 | - | 61.83 |
| baseline | 89.75 | 60.92 | 32.71 | 77.40 | - | 65.20 | baseline | 78.88 | 58.83 | 26.10 | 82.20 | - | 61.50 |
| **Llama-3.2-3B-Instruct** | | | | | | | **Llama-3.2-3B-Instruct** | | | | | | |
| 20_-0.2 | 85.38 | 60.25 | 25.93 | 77.80 | 31.25 | 56.12 | 20_-0.2 | 73.00 | 58.04 | 13.68 | 78.80 | 27.32 | 50.17 |
| 10_-0.2 | 90.31 | 62.08 | 25.03 | 76.40 | 27.69 | 56.30 | 10_-0.2 | 79.12 | 60.21 | 13.32 | 79.40 | 26.66 | 51.74 |
| 20_0.2 | 91.44 | 64.33 | 37.33 | 73.40 | 27.21 | 58.74 | 20_0.2 | 83.69 | 60.25 | 20.58 | 80.60 | 26.09 | 54.20 |
| 10_0.2 | 93.75 | 65.79 | 35.92 | 75.60 | 25.67 | 59.35 | 10_0.2 | 83.69 | 62.08 | 20.77 | 80.40 | 25.94 | 54.58 |
| MI_10 | 91.69 | 63.79 | 35.43 | 73.40 | 27.01 | 58.26 | MI_10 | 82.75 | 60.17 | 21.65 | 79.80 | 25.53 | 53.98 |
| IN_10 | 94.06 | 65.00 | 36.34 | 76.40 | 28.11 | 59.98 | IN_10 | 84.62 | 61.75 | 20.73 | 81.00 | 26.29 | 54.88 |
| baseline | 92.81 | 64.71 | 29.82 | 76.80 | 27.64 | 58.36 | baseline | 84.38 | 63.00 | 17.13 | 80.20 | 26.78 | 54.30 |
| **Qwen2.5-7B** | | | | | | | **Qwen2.5-7B** | | | | | | |
| 20_-0.2 | 95.94 | 58.50 | 32.91 | 75.80 | - | 65.79 | 20_-0.2 | 94.56 | 53.50 | 22.86 | 77.60 | - | 62.13 |
| 10_-0.2 | 96.62 | 59.92 | 32.78 | 75.80 | - | 66.28 | 10_-0.2 | 95.31 | 54.58 | 24.84 | 78.40 | - | 63.28 |
| 20_0.2 | 97.50 | 59.67 | 32.11 | 75.60 | - | 66.22 | 20_0.2 | 95.88 | 54.04 | 23.25 | 79.40 | - | 63.14 |
| 10_0.2 | 97.00 | 59.42 | 32.73 | 75.60 | - | 66.19 | 10_0.2 | 95.50 | 54.08 | 23.11 | 80.00 | - | 63.17 |
| baseline | 96.56 | 60.08 | 33.14 | 76.00 | - | 66.45 | baseline | 96.00 | 54.21 | 23.15 | 79.60 | - | 63.24 |
| **Qwen2.5-7B-Instruct** | | | | | | | **Qwen2.5-7B-Instruct** | | | | | | |
| 20_-0.2 | 96.38 | 62.67 | 46.07 | 76.00 | 40.34 | 64.29 | 20_-0.2 | 94.38 | 55.87 | 35.88 | 78.40 | 33.73 | 59.65 |
| 10_-0.2 | 97.19 | 63.58 | 46.79 | 75.60 | 39.72 | 64.58 | 10_-0.2 | 95.38 | 56.54 | 35.78 | 78.40 | 33.63 | 59.94 |
| 20_0.2 | 97.38 | 65.04 | 45.66 | 75.40 | 41.40 | 64.97 | 20_0.2 | 95.44 | 58.50 | 36.75 | 78.40 | 33.45 | 60.51 |
| 10_0.2 | 98.12 | 64.75 | 46.74 | 76.20 | 39.48 | 65.06 | 10_0.2 | 95.44 | 58.79 | 35.85 | 78.20 | 32.61 | 60.18 |
| MI_10 | 97.94 | 65.08 | 46.91 | 76.80 | 40.18 | 65.38 | MI_10 | 94.94 | 57.83 | 35.87 | 78.20 | 31.58 | 59.68 |
| IN_10 | 98.06 | 64.75 | 47.68 | 76.00 | 40.79 | 65.46 | IN_10 | 95.25 | 57.71 | 35.29 | 77.40 | 32.92 | 59.71 |
| baseline | 98.00 | 64.83 | 47.15 | 75.80 | 40.93 | 65.34 | baseline | 95.25 | 57.71 | 36.56 | 77.40 | 31.92 | 59.77 |
| **Ministral-8B-Instruct-2410** | | | | | | | **Ministral-8B-Instruct-2410** | | | | | | |
| 20_-0.2 | 97.19 | 65.29 | 49.82 | 72.80 | 35.26 | 64.07 | 20_-0.2 | 94.62 | 61.79 | 31.31 | 77.20 | 33.59 | 59.70 |
| 10_-0.2 | 97.75 | 65.12 | 47.68 | 74.20 | 34.01 | 63.75 | 10_-0.2 | 94.56 | 62.17 | 29.74 | 78.80 | 33.17 | 59.69 |
| 20_0.2 | 97.06 | 66.00 | 49.30 | 76.60 | 32.93 | 64.38 | 20_0.2 | 93.81 | 63.46 | 38.86 | 79.40 | 29.00 | 60.91 |
| 10_0.2 | 96.94 | 65.58 | 49.17 | 75.80 | 32.39 | 63.98 | 10_0.2 | 93.81 | 63.87 | 36.69 | 79.60 | 28.74 | 60.54 |
| MI_10 | 97.25 | 66.62 | 49.12 | 76.40 | 33.49 | 64.58 | MI_10 | 94.56 | 63.33 | 39.01 | 79.60 | 34.05 | 62.11 |
| IN_10 | 97.12 | 66.75 | 50.77 | 74.80 | 34.33 | 64.76 | IN_10 | 93.81 | 62.79 | 37.44 | 77.20 | 34.60 | 61.17 |
| baseline | 97.75 | 66.00 | 48.55 | 76.80 | 32.78 | 64.38 | baseline | 94.75 | 63.58 | 33.68 | 79.00 | 31.56 | 60.51 |

Table 2: HELMET benchmark results under 16k (left) and 32k (right) context. Green indicates better than the baseline; red indicates worse than the baseline. "10_0.2" means intervention top-10 heads with $\alpha = 0.2$. "IN_10" means automated magnitude control with the IN objective with top-10 heads.

the base model does not have good task alignment and can benefit from applying focus direction. Second, if there is an unusual sink contextual score, focus directions could help to achieve a better task alignment by paying the right amount of attention to the contexts. For example, for the TREC Coarse task under 8k contexts, the Llama-instruction model has a sink contextual score of 0.535, higher than the average score under 8k contexts of 0.297. As such, the LLM may not pay enough attention to the contexts. A positive focus direction helps the performance improve from 69% to 75%.

**Most of the tasks could be improved by either a positive or a negative focus direction.** Table 2 shows the category-based average performance of each task under 16k and 32k contexts. We found that 32 of the 46 task categories could have performance improvement by either positive or negative focus directions. This indicates that the focus direction generalizes to most long-context tasks. This also confirms that the proper amount of attention activation is needed for optimal task performance. When an LLM exhibits excessive attention activation, a negative focus direction may help suppress irrelevant information. Conversely, when attention activation is insufficient, a positive focus direction can enhance attention to relevant contexts.

**Focus direction improves the overall performance of poorly aligned LLMs.** We also show the overall average performance of all the tasks. We found that focus direction could improve the performance of 5 of 5 LLMs on 32k contexts and 3 of 5 LLMs on 16k contexts. We also check the standard deviation of the sink contextual scores of all the tasks for each LLM (Table 13 and 14). We consider the LLMs with higher standard deviation poorly aligned since they do not have a consistent attention behavior under the same length of context. Based on this, we consider that Qwen and LLama are more poorly aligned than Ministral. And over the performance of different tasks ranging from 8k to 128k contexts, Qwen and LLama have more improvement than Ministral with the focus directions. We conclude that focus directions are likely to improve poorly aligned LLMs.

**Proper magnitude control can improve the overall performance of LLMs.** Among the 30 task categories evaluated under 16k and 32k contexts, as shown in Table 2, our automatic magnitude control (IN objective) improves or maintains the performance in 19 cases, demonstrating its effectiveness. We also note the limitation of the focus directions in cross-domain generalization. In such cases, a proper magnitude means a conservative intervention (e.g., top-10 heads, $\alpha = 0.1$). In comparison, in the in-domain case in Section 3, an aggressive intervention (e.g., top-20 heads, $\alpha = 0.3$) could be applied to further enhance performance. To further enhance the performance, task-dependent focus directions are needed. We further discuss this in Section 5.

## 5 DISCUSSION

**Contextual heads vs. retrieval heads.** A similar type of attention head with contextual heads is retrieval heads Wu et al. (2024). Retrieval heads are the attention heads used for copying tokens from the input to the output. We found that contextual heads are different from retrieval heads in the following aspects. 1) Location: As shown in Figure 9, retrieval heads universally exist in different layers, while contextual heads are mainly located in the middle and late layers. Among the top 20 retrieval heads and contextual heads, only 5 overlap in the Llama-3.2-3B-Instruct model. 2) Function: retrieval heads focus on explicit copy tokens from the input to the output, while contextual heads control the overall attention of LLMs.

**Focus directions may be task dependent.** While we verify the existence of the focus direction, we do not consider that we locate the "optimal" focus direction for every task. Instead, we consider that the focus direction may be task-dependent. In other words, each task may have a different definition of relevant contexts and may have its corresponding focus directions. Given such task-dependent focus directions, more aggressive interventions could be applied to further improve the performance. In addition, given optimal task focus directions, the overall amount of attention activation may converge across tasks that share the same context length. We leave these as future work.

**Border impact of contextual heads and focus directions.** We consider that the focus direction may have the following applications: 1) Focus directions may be an alternative approach for parameter-efficient fine-tuning Xu et al. (2023) for adapting long-context language models for different tasks. 2) Focus directions may serve as a "switch" to control the LLM's use of contextual or internal knowledge, addressing knowledge conflicts Xu et al. (2024).

## 6 RELATED WORK

**Long context LLMs and evaluation.** Advanced long-context LLMs now can accommodate 128k or more tokens in their context, including property models like GPT-4, Gemini, and Claude and open-source models like Llama 3.1 Dubey et al. (2024), Ministral and Qwen2.5 Yang et al. (2024). Such models enable various applications, such as long context QA Wang et al. (2024); Karpinska et al. (2024), in-context learning Li et al. (2024c); Agarwal et al. (2025); Bertsch et al. (2024), summarization Chang et al. (2023); Kim et al. (2024), and retrieval-augmented generation Lee et al. (2024). For evaluation, early works mainly focus on the synthetic tasks Hsieh et al. (2024); Liu et al. (2024); Tay et al. (2020), such as the needle in the haystack, which may not well measure the LLM performance in the real world. Recent work has focused more on diverse and real-world settings, such as RAG Lee et al. (2024), in-context learning Li et al. (2024c), and reasoning Zhou et al. (2025).

**Mechanistic interpretability on attention heads and activation steering.** Our contextual heads relate to the recent work that discovers functional attention heads in LLMs, such as heads related to retrieval Wu et al. (2024), in-context learning Olsson et al. (2022); Yin & Steinhardt (2025); Ren et al. (2024), safety Chen et al. (2024), and knowledge conflicts Jin et al. (2024); Shi et al. (2024). Our focus direction is related to the activation steering work, which could use a directional vector to control the LLMs' behavior, such as truthfulness Li et al. (2024b), sentiment Han et al. (2023), and refusal Arditi et al. (2024).

## 7 CONCLUSION

In this paper, we identify the contextual heads that control the overall attention of LLMs to contexts and focus directions on these heads, which can make LLMs pay more attention to relevant contexts. We first propose a contextual scoring method to identify the contextual heads. Then, we demonstrate that insufficient attention to the relevant context in these heads is the cause of LLM distraction. Moreover, we identify focus directions that could redirect the attention of contextual heads from the attention sink to the relevant contexts, thereby mitigating distraction. We additionally introduce an automatic magnitude control method to control the strength of focus directions and make them applicable to the real world. We further study the effect of focus directions on the real-world long context benchmark and find that focus directions could help mitigate poor task alignment. At last, we discuss the potential border impact of focus directions for long-context LLM alignment.

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

# A EXPERIMENT DETAILS

## A.1 DETAILS OF EXPERIMENT ON HELMET BENCHMARK

We used the same settings as the HELMET benchmark. For metrics, we used the substring exact match for all the retrieval-augmented generation and synthetic recall tasks, NDCG@10 for the passage re-ranking task, and accuracy for all many-shot in-context learning tasks, ROUGE F1 for Infbench QA, and accuracy for the Infbench MC. We exclude other tasks that require model-based evaluation.

# B ADDITIONAL RESULTS

We provide additional results for the Qwen and Ministral models in Figure 5, 6 (contextual head locations), Figure 7, 8 (EM accuracy of test split of the Multi-Document Question Answering data).

# C    AUTOMATED MAGNITUDE CONTROL

## C.1    MOTIVATIONS

Different samples may require different amounts of attention activation to achieve optimal performance. Additionally, it is a challenge to determine a fixed magnitude for real-world applications. To address this, we introduce a method for automated magnitude control. We consider two types of training objectives for automated magnitude control.

**Median Intervention (MI).** We first observe, in the case of intervention with top-10 heads, 99% of the sample in our training set has a consecutive range of magnitudes that could result in a correct response. We thus consider the most robust magnitude for such samples to be the median of that consecutive range. We visualize the distribution of the median values of the Llama-3.2-3B instruction model (top-10 heads) in Figure 10. We use such median values as the training objectives.

**Intervention when Necessary (IN).** We also observe that among the samples that can produce a correct response with or without intervention ($\alpha = [-1, 1]$), 61% of those samples can produce a correct response without intervention (i.e., $\alpha = 0$). We thus consider another strategy that only applies minimal intervention to make the response correct. To this end, we define the training objective as the magnitude closest to 0 for a correct response. The distribution of such magnitudes is shown in Figure 11.

## C.2    METHOD

Our method to achieve automated magnitude control consists of three steps: magnitude sampling, magnitude controller training, and inference.

**Magnitude sampling.** Given a predefined top-$k$ heads, we first sample different magnitudes and record the corresponding prompt, correct responses, and magnitude tuples $(P, R, \alpha)$. In our case, this can be achieved using the training set defined in §2.1. We sample top-10 and top-20 contextual heads with magnitudes ranging from -1 to 1, incrementing by 0.1 intervals. For the MI objective, for each prompt $P$, we only keep the tuple with the median $\alpha$ for training. For the IN objective, we only keep the tuple with the $\alpha$ closest to 0 for training.

**Magnitude controller training.** We used linear layers as our magnitude controller. At each transformer attention layer, these controllers receive the same hidden states used by the key and query projections as input, and produce a single scalar value per token for every attention head in that layer. We use the KL divergence loss to train the magnitude controller. Specifically, we first use a fixed teacher model to obtain the output probability distribution for each token given the input of $[P, R]$ under the intervention strength $\alpha$. Then, we use the probability distribution to train the student model using the KL divergence loss. The student model is also fixed, except that the magnitude controllers are being trained. We used a learning rate of $10^{-4}$ and trained for 10 epochs.

**Inference.** Given the trained magnitude controllers, we can now obtain a dynamic magnitude for each head at each token. To avoid the unnecessary perturbation of magnitude across tokens during inference. We first average the magnitudes across tokens for each head. This averaged value is then fixed and used consistently during text generation.

## C.3    RESULTS

We first discuss the in-domain result of the Multi-Document Question Answering data. We consider both top-10 and top-20 heads. As shown in Table 4, all three LLMs have a performance gain with automated magnitude control, demonstrating the effectiveness of the proposed method. Also, compared to the IN objective, the MI objective results in better performance in the in-domain testing set. This suggests, in the case of an in-domain application, the MI objective is preferred. We also consider that the MI objective introduces a stronger intervention than the IN objective. Thus, in the case of an in-domain application, an aggressive intervention is preferred.

We then discuss the cross-domain result on the HELMET benchmark. We first observed, among 15 cases of the overall average of three LLMs in 8-128k contexts, that the IN objective outperformed the MI objective in 12 cases. This suggests that, in cross-domain applications, a conservative intervention

is preferred. This also suggests that the focus directions we found can generalize across domains, but in limited ways. Such directions are likely to help when the initial amount of attention activation is abnormal, but less likely to further improve the performance when the amount of attention activation is close to optimal.

# D MISCELLANEOUS

## D.1 LLM USAGE

LLMs were used for polishing writing (i.e., correcting grammatical errors and enhancing writing clarity).

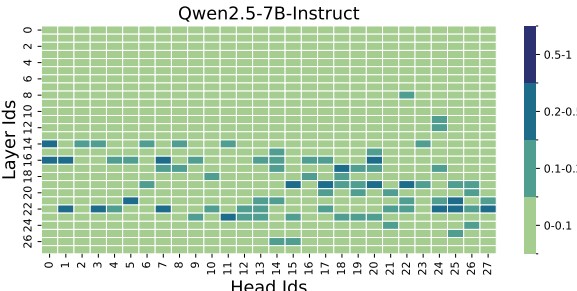

Figure 5: Location of the contextual heads of Qwen2.5-7B-Instruct.

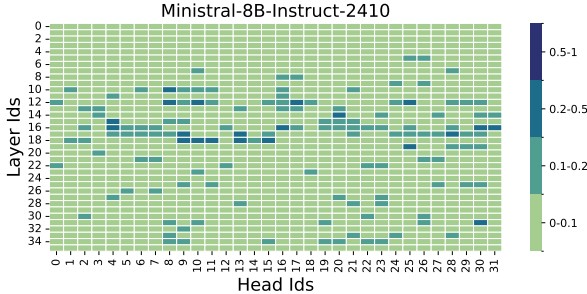

Figure 6: Location of the contextual heads Ministral-8B-Instruct-2410.

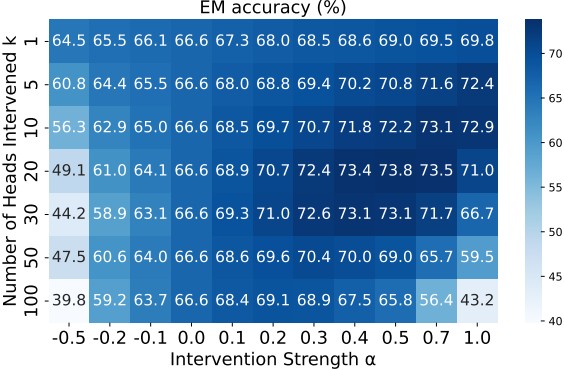

Figure 7: EM accuracy of different top-$k$ heads and $\alpha$ of Qwen2.5-7B-Instruct.

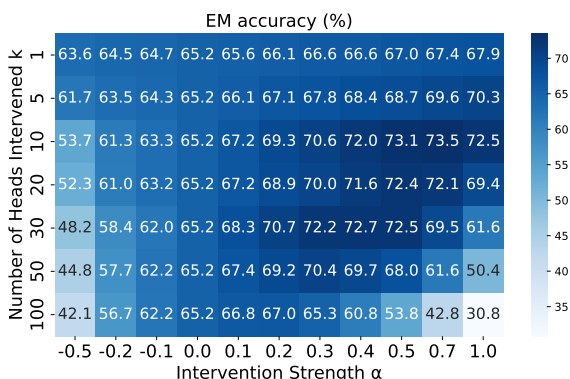

Figure 8: EM accuracy of different top-$k$ heads and $\alpha$ of Ministral-8B-Instruct-2410.

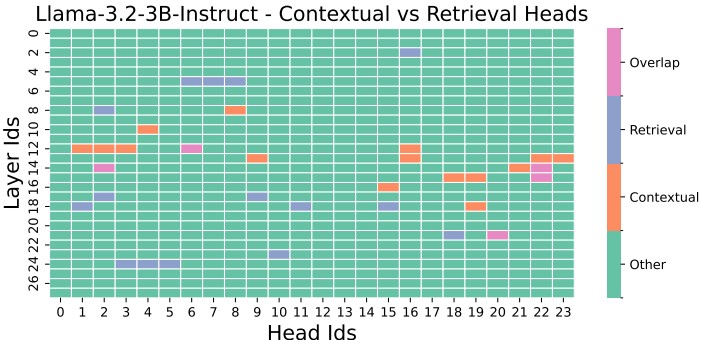

Figure 9: The location of top-20 contextual head vs. top-20 retrieval heads.

| Heads | R | IR | IR max | Sink | R Gold | Sink Gold | Heads | R | IR | IR max | Sink |
|---|---|---|---|---|---|---|---|---|---|---|---|
| $\alpha = -0.2$ | | | | | | | $\alpha = 0.4$ | | | | |
| (13, 23) | 0.10 (-0.11) | 0.43 (-0.08) | 0.12 (-0.04) | 0.30 (+0.19) | 0.30 (-0.26) | 0.42 (+0.23) | (13, 23) | 0.40784 | 0.46287 | 0.21073 | 0.00626 |
| (12, 1) | 0.12 (-0.08) | 0.56 (-0.01) | 0.14 (-0.02) | 0.14 (+0.06) | 0.45 (-0.19) | 0.31 (+0.15) | (12, 1) | 0.40324 | 0.50762 | 0.21712 | 0.01094 |
| (15, 18) | 0.09 (-0.11) | 0.35 (-0.08) | 0.10 (-0.04) | 0.44 (+0.18) | 0.25 (-0.26) | 0.55 (+0.24) | (15, 18) | 0.37799 | 0.48941 | 0.20475 | 0.04639 |
| (15, 22) | 0.09 (-0.10) | 0.31 (-0.08) | 0.10 (-0.04) | 0.40 (+0.16) | 0.26 (-0.23) | 0.49 (+0.21) | (15, 22) | 0.36006 | 0.48624 | 0.20482 | 0.04563 |
| (14, 2) | 0.07 (-0.12) | 0.21 (-0.13) | 0.08 (-0.05) | 0.56 (+0.27) | 0.12 (-0.23) | 0.72 (+0.26) | (14, 2) | 0.44528 | 0.45260 | 0.23089 | 0.01582 |
| $\alpha = 0.2$ | | | | | | | $\alpha = 0.5$ | | | | |
| (13, 23) | 0.31 (+0.11) | 0.51 (-0.00) | 0.18 (+0.02) | 0.03 (-0.08) | 0.75 (+0.20) | 0.06 (-0.13) | (13, 23) | 0.41196 | 0.47068 | 0.23630 | 0.00324 |
| (12, 1) | 0.29 (+0.09) | 0.55 (-0.02) | 0.18 (+0.02) | 0.04 (-0.04) | 0.78 (+0.14) | 0.06 (-0.09) | (12, 1) | 0.41427 | 0.51336 | 0.24182 | 0.00608 |
| (15, 18) | 0.32 (+0.12) | 0.45 (+0.03) | 0.17 (+0.04) | 0.12 (-0.13) | 0.74 (+0.24) | 0.13 (-0.18) | (15, 18) | 0.38408 | 0.51542 | 0.22953 | 0.02423 |
| (15, 22) | 0.31 (+0.12) | 0.44 (+0.05) | 0.18 (+0.04) | 0.12 (-0.13) | 0.69 (+0.21) | 0.12 (-0.16) | (15, 22) | 0.36751 | 0.52062 | 0.23100 | 0.02447 |
| (14, 2) | 0.32 (+0.13) | 0.42 (+0.08) | 0.17 (+0.04) | 0.10 (-0.19) | 0.64 (+0.29) | 0.19 (-0.27) | (14, 2) | 0.45509 | 0.47693 | 0.26239 | 0.00799 |
| $\alpha = 0.5$ | | | | | | | $\alpha = 1.0$ | | | | |
| (13, 23) | 0.45 (+0.24) | 0.46 (-0.05) | 0.21 (+0.05) | 0.00 (-0.10) | 0.90 (+0.34) | 0.00 (-0.18) | (13, 23) | 0.34991 | 0.55895 | 0.34943 | 0.00008 |
| (12, 1) | 0.41 (+0.21) | 0.51 (-0.06) | 0.21 (+0.05) | 0.01 (-0.07) | 0.90 (+0.26) | 0.01 (-0.14) | (12, 1) | 0.36854 | 0.59642 | 0.33347 | 0.00010 |
| (15, 18) | 0.47 (+0.27) | 0.46 (+0.03) | 0.22 (+0.08) | 0.02 (-0.24) | 0.93 (+0.42) | 0.02 (-0.30) | (15, 18) | 0.33626 | 0.61708 | 0.32412 | 0.00017 |
| (15, 22) | 0.46 (+0.27) | 0.46 (+0.07) | 0.22 (+0.08) | 0.01 (-0.23) | 0.89 (+0.41) | 0.01 (-0.27) | (15, 22) | 0.31526 | 0.64934 | 0.32020 | 0.00054 |
| (14, 2) | 0.47 (+0.29) | 0.44 (+0.10) | 0.22 (+0.09) | 0.01 (-0.29) | 0.91 (+0.57) | 0.02 (-0.44) | (14, 2) | 0.38478 | 0.58332 | 0.36628 | 0.00018 |

Table 3: **Left:** Contextual scores of top-5 contextual heads when top-5 heads are intervened. The value in the "()" represents the difference compared to the result without intervention in Table 1. **Right:** Contextual scores of top-5 contextual heads when top-20 heads are intervened.

| Model/Top-k heads | Baseline | MI_10 | MI_20 | IN_10 | IN_20 |
|---|---|---|---|---|---|
| Llama-3.2-3B-Instruct | 59.4 | 64.9 | 66.4 | 62.3 | 62.1 |
| Qwen2.5-7B-Instruct | 66.6 | 69.1 | 71.7 | 67.2 | 68.6 |
| Ministral-8B-Instruct-2410 | 65.2 | 67.8 | 69.0 | 65.8 | 67.0 |

Table 4: Automated magnitude control result (EM accuracy %) on testing split of the Multi-Document Question Answering data.

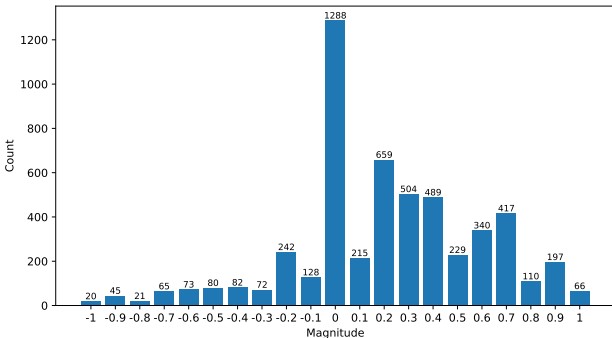

Figure 10: Magnitude distribution of Llama-3.2-3B-Instruct model with median intervention objective (top-10 contextual heads).

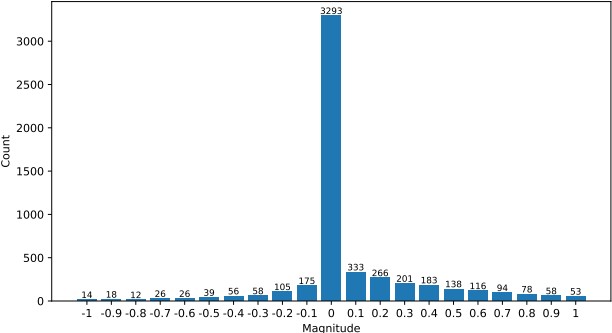

Figure 11: Magnitude distribution of Llama-3.2-3B-Instruct model with intervention when necessary objective (top-10 contextual heads).

| model | Recall MK Needle | MK UUID | MV | JSON KV | NQ | RAG TriviaQA | HotpotQA | PopQA | Re-ranking MS MARCO | TREC Coarse | TREC Fine | ICL BANKING77 | CLINC150 | NLU | Long QA Infbench QA | Infbench MC |
|---|---|---|---|---|---|---|---|---|---|---|---|---|---|---|---|---|
| **Llama-3.2-3B** | | | | | | | | | | | | | | | | |
| 20_-0.2 | 90.00 | 89.00 | 53.00 | 98.00 | 48.67 | 82.33 | 48.33 | 52.67 | 38.15 | 58.00 | 27.00 | 66.00 | 73.00 | 77.00 | - | - |
| 10_-0.2 | 99.00 | 95.00 | 60.50 | 100.00 | 50.67 | 82.33 | 50.67 | 54.00 | 42.31 | 59.00 | 28.00 | 66.00 | 75.00 | 78.00 | - | - |
| 20_0.2 | 98.00 | 97.00 | 64.50 | 98.00 | 54.50 | 79.33 | 55.67 | 56.17 | 46.21 | 67.00 | 34.00 | 68.00 | 78.00 | 82.00 | - | - |
| 10_0.2 | 99.00 | 99.00 | 67.25 | 98.00 | 54.67 | 81.00 | 56.00 | 56.83 | 42.41 | 67.00 | 35.00 | 69.00 | 78.00 | 81.00 | - | - |
| baseline | 99.00 | 99.00 | 65.50 | 99.00 | 54.50 | 82.17 | 52.67 | 56.50 | 43.52 | 61.00 | 33.00 | 69.00 | 77.00 | 82.00 | - | - |
| **Llama-3.2-3B-Instruct** | | | | | | | | | | | | | | | | |
| 20_-0.2 | 100.00 | 90.00 | 99.00 | 88.00 | 51.17 | 83.00 | 54.00 | 56.83 | 35.19 | 68.00 | 37.00 | 77.00 | 72.00 | 76.00 | 14.47 | 36.00 |
| 10_-0.2 | 100.00 | 98.00 | 99.00 | 96.00 | 51.50 | 82.17 | 57.67 | 59.83 | 36.71 | 67.00 | 34.00 | 76.00 | 70.00 | 79.00 | 15.99 | 37.00 |
| 20_0.2 | 99.00 | 98.00 | 99.75 | 97.00 | 56.50 | 82.67 | 62.67 | 62.00 | 48.37 | 75.00 | 33.00 | 69.00 | 69.00 | 79.00 | 16.63 | 29.00 |
| 10_0.2 | 99.00 | 97.00 | 99.00 | 98.00 | 58.00 | 85.17 | 62.67 | 60.67 | 46.90 | 75.00 | 33.00 | 72.00 | 72.00 | 79.00 | 13.88 | 27.00 |
| MI_10 | 97.00 | 98.00 | 100.00 | 96.00 | 57.67 | 83.83 | 62.67 | 60.83 | 46.37 | 75.00 | 34.00 | 73.00 | 71.00 | 78.00 | 14.91 | 27.00 |
| IN_10 | 99.00 | 99.00 | 98.75 | 97.00 | 56.17 | 84.33 | 62.67 | 59.33 | 47.15 | 70.00 | 32.00 | 73.00 | 70.00 | 79.00 | 14.83 | 29.00 |
| baseline | 100.00 | 99.00 | 99.00 | 99.00 | 55.33 | 84.83 | 62.00 | 59.17 | 41.68 | 69.00 | 35.00 | 72.00 | 71.00 | 80.00 | 15.48 | 31.00 |
| **Qwen2.5-7B** | | | | | | | | | | | | | | | | |
| 20_-0.2 | 99.00 | 99.00 | 95.00 | 98.00 | 51.83 | 79.00 | 61.00 | 53.17 | 48.67 | 82.00 | 43.00 | 66.00 | 75.00 | 75.00 | - | - |
| 10_-0.2 | 100.00 | 99.00 | 95.50 | 99.00 | 52.50 | 79.33 | 62.00 | 53.83 | 48.55 | 82.00 | 45.00 | 66.00 | 76.00 | 73.00 | - | - |
| 20_0.2 | 100.00 | 96.00 | 95.25 | 100.00 | 50.17 | 78.50 | 64.67 | 57.50 | 44.94 | 81.00 | 41.00 | 66.00 | 76.00 | 74.00 | - | - |
| 10_0.2 | 100.00 | 96.00 | 95.25 | 100.00 | 51.33 | 79.17 | 65.00 | 56.17 | 46.52 | 83.00 | 42.00 | 70.00 | 75.00 | 74.00 | - | - |
| baseline | 100.00 | 99.00 | 95.00 | 100.00 | 51.50 | 79.83 | 64.00 | 55.83 | 47.44 | 83.00 | 43.00 | 68.00 | 75.00 | 74.00 | - | - |
| **Qwen2.5-7B-Instruct** | | | | | | | | | | | | | | | | |
| 20_-0.2 | 100.00 | 99.00 | 98.00 | 97.00 | 53.17 | 78.67 | 58.00 | 58.50 | 58.53 | 82.00 | 42.00 | 76.00 | 80.00 | 75.00 | 22.17 | 42.00 |
| 10_-0.2 | 100.00 | 98.00 | 98.50 | 98.00 | 53.83 | 79.17 | 58.67 | 58.67 | 56.51 | 82.00 | 44.00 | 73.00 | 80.00 | 76.00 | 22.45 | 42.00 |
| 20_0.2 | 100.00 | 97.00 | 98.25 | 98.00 | 52.50 | 82.83 | 62.67 | 63.17 | 56.25 | 81.00 | 41.00 | 70.00 | 79.00 | 76.00 | 25.20 | 40.00 |
| 10_0.2 | 100.00 | 97.00 | 98.25 | 100.00 | 54.00 | 82.33 | 61.67 | 62.50 | 56.98 | 81.00 | 40.00 | 73.00 | 80.00 | 77.00 | 25.54 | 41.00 |
| MI_10 | 100.00 | 98.00 | 98.50 | 100.00 | 53.67 | 82.17 | 61.33 | 62.00 | 57.33 | 83.00 | 41.00 | 72.00 | 81.00 | 77.00 | 23.90 | 41.00 |
| IN_10 | 100.00 | 98.00 | 98.50 | 100.00 | 54.83 | 82.17 | 60.67 | 61.17 | 57.76 | 83.00 | 41.00 | 72.00 | 81.00 | 76.00 | 25.61 | 40.00 |
| baseline | 100.00 | 98.00 | 98.00 | 98.75 | 54.33 | 81.83 | 59.67 | 61.17 | 57.54 | 83.00 | 41.00 | 73.00 | 81.00 | 76.00 | 24.12 | 40.00 |
| **Ministral-8B-Instruct-2410** | | | | | | | | | | | | | | | | |
| 20_-0.2 | 100.00 | 100.00 | 93.75 | 100.00 | 57.00 | 87.00 | 65.33 | 54.83 | 65.00 | 73.00 | 38.00 | 83.00 | 86.00 | 78.00 | 26.30 | 36.00 |
| 10_-0.2 | 100.00 | 100.00 | 94.75 | 100.00 | 56.33 | 87.00 | 64.33 | 54.17 | 64.95 | 73.00 | 38.00 | 83.00 | 86.00 | 79.00 | 25.87 | 38.00 |
| 20_0.2 | 100.00 | 100.00 | 95.50 | 100.00 | 53.83 | 87.83 | 68.00 | 59.17 | 66.88 | 74.00 | 41.00 | 82.00 | 84.00 | 79.00 | 24.04 | 35.00 |
| 10_0.2 | 100.00 | 100.00 | 95.25 | 100.00 | 53.83 | 87.50 | 67.00 | 59.33 | 65.80 | 74.00 | 38.00 | 83.00 | 83.00 | 80.00 | 23.39 | 33.00 |
| MI_10 | 100.00 | 100.00 | 95.25 | 100.00 | 54.00 | 87.17 | 67.67 | 58.83 | 67.59 | 75.00 | 40.00 | 83.00 | 83.00 | 79.00 | 24.89 | 36.00 |
| IN_10 | 100.00 | 100.00 | 95.50 | 100.00 | 55.67 | 87.17 | 68.00 | 58.00 | 68.42 | 74.00 | 39.00 | 81.00 | 84.00 | 79.00 | 25.36 | 36.00 |
| baseline | 100.00 | 100.00 | 96.00 | 100.00 | 54.83 | 86.50 | 67.67 | 58.67 | 66.32 | 74.00 | 37.00 | 84.00 | 85.00 | 79.00 | 27.09 | 37.00 |

Table 5: Results of HELMET benchmark under 8k context.

| model | Recall | | | | RAG | | | | Re-ranking | | | ICL | | | Long QA | |
| | MK Needle | MK UUID | MV | JSON KV | NQ | TriviaQA | HotpotQA | PopQA | MS MARCO | TREC Coarse | TREC Fine | BANKING77 | CLINC150 | NLU | Infbench QA | Infbench MC |
|---|---|---|---|---|---|---|---|---|---|---|---|---|---|---|---|---|
| **Llama-3.2-3B** | | | | | | | | | | | | | | | | |
| 20_-0.2 | 94.00 | 73.00 | 66.75 | 95.00 | 47.00 | 86.33 | 43.67 | 48.00 | 30.15 | 82.00 | 44.00 | 83.00 | 87.00 | 80.00 | - | - |
| 10_-0.2 | 99.00 | 84.00 | 73.75 | 96.00 | 50.83 | 87.00 | 46.33 | 50.50 | 32.46 | 82.00 | 42.00 | 83.00 | 90.00 | 79.00 | - | - |
| 20_0.2 | 98.00 | 91.00 | 66.25 | 96.00 | 52.50 | 82.50 | 51.33 | 54.83 | 31.44 | 83.00 | 42.00 | 87.00 | 90.00 | 81.00 | - | - |
| 10_0.2 | 99.00 | 90.00 | 74.25 | 96.00 | 53.17 | 84.17 | 50.00 | 54.17 | 31.54 | 84.00 | 42.00 | 89.00 | 89.00 | 83.00 | - | - |
| baseline | 99.00 | 90.00 | 74.00 | 96.00 | 55.00 | 85.83 | 50.00 | 52.83 | 32.71 | 82.00 | 45.00 | 89.00 | 89.00 | 82.00 | - | - |
| **Llama-3.2-3B-Instruct** | | | | | | | | | | | | | | | | |
| 20_-0.2 | 100.00 | 55.00 | 99.50 | 87.00 | 50.67 | 84.00 | 50.33 | 56.00 | 25.93 | 81.00 | 52.00 | 86.00 | 87.00 | 83.00 | 21.51 | 41.00 |
| 10_-0.2 | 99.00 | 74.00 | 99.25 | 89.00 | 53.67 | 84.17 | 52.33 | 58.17 | 25.03 | 80.00 | 46.00 | 84.00 | 88.00 | 84.00 | 20.38 | 35.00 |
| 20_0.2 | 97.00 | 81.00 | 99.75 | 88.00 | 54.33 | 85.00 | 58.33 | 59.67 | 37.33 | 79.00 | 34.00 | 84.00 | 87.00 | 83.00 | 21.42 | 33.00 |
| 10_0.2 | 99.00 | 83.00 | 100.00 | 93.00 | 56.00 | 86.83 | 61.00 | 59.33 | 35.92 | 81.00 | 39.00 | 88.00 | 89.00 | 81.00 | 19.35 | 32.00 |
| MI_10 | 98.00 | 76.00 | 99.75 | 93.00 | 53.17 | 85.17 | 60.33 | 56.50 | 35.43 | 78.00 | 36.00 | 85.00 | 87.00 | 81.00 | 24.02 | 30.00 |
| IN_10 | 100.00 | 83.00 | 99.25 | 94.00 | 56.17 | 86.33 | 59.00 | 58.50 | 36.34 | 82.00 | 40.00 | 87.00 | 89.00 | 84.00 | 23.22 | 33.00 |
| baseline | 99.00 | 82.00 | 99.25 | 91.00 | 54.67 | 86.00 | 58.67 | 59.50 | 29.82 | 83.00 | 41.00 | 85.00 | 90.00 | 85.00 | 24.28 | 31.00 |
| **Qwen2.5-7B** | | | | | | | | | | | | | | | | |
| 20_-0.2 | 96.00 | 96.00 | 94.75 | 97.00 | 50.50 | 78.00 | 58.67 | 46.83 | 32.91 | 86.00 | 46.00 | 84.00 | 89.00 | 74.00 | - | - |
| 10_-0.2 | 97.00 | 96.00 | 95.50 | 98.00 | 51.33 | 80.00 | 59.33 | 49.00 | 32.78 | 87.00 | 46.00 | 83.00 | 88.00 | 75.00 | - | - |
| 20_0.2 | 100.00 | 95.00 | 96.00 | 99.00 | 49.67 | 78.33 | 59.67 | 51.00 | 32.11 | 89.00 | 47.00 | 79.00 | 86.00 | 77.00 | - | - |
| 10_0.2 | 99.00 | 94.00 | 96.00 | 99.00 | 49.33 | 77.33 | 58.67 | 52.33 | 32.73 | 88.00 | 45.00 | 81.00 | 88.00 | 76.00 | - | - |
| baseline | 97.00 | 96.00 | 95.25 | 98.00 | 50.17 | 79.83 | 59.67 | 50.67 | 33.14 | 88.00 | 45.00 | 83.00 | 89.00 | 75.00 | - | - |
| **Qwen2.5-7B-Instruct** | | | | | | | | | | | | | | | | |
| 20_-0.2 | 99.00 | 99.00 | 96.50 | 91.00 | 51.50 | 83.33 | 57.33 | 58.50 | 46.07 | 86.00 | 44.00 | 86.00 | 89.00 | 75.00 | 30.67 | 50.00 |
| 10_-0.2 | 99.00 | 99.00 | 96.75 | 94.00 | 53.50 | 82.67 | 59.67 | 58.50 | 46.79 | 85.00 | 44.00 | 84.00 | 89.00 | 76.00 | 30.45 | 49.00 |
| 20_0.2 | 99.00 | 97.00 | 96.50 | 97.00 | 53.67 | 83.33 | 63.33 | 59.83 | 45.66 | 83.00 | 45.00 | 83.00 | 89.00 | 77.00 | 30.80 | 52.00 |
| 10_0.2 | 99.00 | 99.00 | 97.50 | 97.00 | 54.50 | 83.17 | 61.33 | 60.00 | 46.74 | 84.00 | 46.00 | 85.00 | 89.00 | 77.00 | 29.97 | 49.00 |
| MI_10 | 99.00 | 99.00 | 96.75 | 97.00 | 56.17 | 83.50 | 61.00 | 59.67 | 46.91 | 85.00 | 44.00 | 86.00 | 91.00 | 78.00 | 31.37 | 49.00 |
| IN_10 | 99.00 | 99.00 | 97.25 | 97.00 | 55.50 | 83.33 | 60.00 | 60.17 | 47.68 | 84.00 | 44.00 | 86.00 | 89.00 | 77.00 | 31.59 | 50.00 |
| baseline | 99.00 | 99.00 | 97.00 | 97.00 | 54.67 | 83.50 | 61.33 | 59.83 | 47.15 | 83.00 | 45.00 | 86.00 | 89.00 | 76.00 | 31.87 | 50.00 |
| **Ministral-8B-Instruct-2410** | | | | | | | | | | | | | | | | |
| 20_-0.2 | 100.00 | 100.00 | 90.75 | 98.00 | 56.00 | 88.33 | 63.00 | 53.83 | 49.82 | 84.00 | 33.00 | 87.00 | 90.00 | 70.00 | 30.53 | 40.00 |
| 10_-0.2 | 100.00 | 100.00 | 92.00 | 99.00 | 55.33 | 88.33 | 62.67 | 54.17 | 47.68 | 86.00 | 34.00 | 86.00 | 90.00 | 75.00 | 30.03 | 38.00 |
| 20_0.2 | 100.00 | 100.00 | 89.25 | 99.00 | 53.83 | 89.17 | 64.33 | 56.67 | 49.30 | 89.00 | 35.00 | 88.00 | 89.00 | 82.00 | 26.85 | 39.00 |
| 10_0.2 | 100.00 | 100.00 | 88.75 | 99.00 | 53.67 | 88.83 | 64.00 | 55.83 | 49.17 | 87.00 | 32.00 | 88.00 | 91.00 | 81.00 | 27.78 | 37.00 |
| MI_10 | 100.00 | 100.00 | 91.00 | 98.00 | 55.83 | 88.67 | 65.67 | 56.33 | 49.12 | 87.00 | 35.00 | 88.00 | 92.00 | 80.00 | 28.98 | 38.00 |
| IN_10 | 100.00 | 100.00 | 90.50 | 98.00 | 57.17 | 88.83 | 65.00 | 56.00 | 50.77 | 87.00 | 32.00 | 88.00 | 91.00 | 76.00 | 28.66 | 40.00 |
| baseline | 100.00 | 100.00 | 92.00 | 99.00 | 55.33 | 89.00 | 63.67 | 56.00 | 48.55 | 88.00 | 35.00 | 89.00 | 91.00 | 81.00 | 29.57 | 36.00 |

Table 6: Results of HELMET benchmark under 16k context.

| model | Recall | | | | RAG | | | | Re-ranking | | | ICL | | | Long QA | |
| | MK Needle | MK UUID | MV | JSON KV | NQ | TriviaQA | HotpotQA | PopQA | MS MARCO | TREC Coarse | TREC Fine | BANKING77 | CLINC150 | NLU | Infbench QA | Infbench MC |
|---|---|---|---|---|---|---|---|---|---|---|---|---|---|---|---|---|
| **Llama-3.2-3B** | | | | | | | | | | | | | | | | |
| 20_-0.2 | 88.00 | 43.00 | 54.00 | 79.00 | 45.33 | 84.67 | 43.00 | 46.83 | 29.22 | 87.00 | 59.00 | 90.00 | 89.00 | 86.00 | - | - |
| 10_-0.2 | 95.00 | 54.00 | 56.25 | 90.00 | 44.17 | 86.83 | 46.00 | 49.33 | 26.73 | 86.00 | 62.00 | 92.00 | 89.00 | 86.00 | - | - |
| 20_0.2 | 98.00 | 77.00 | 60.00 | 91.00 | 47.50 | 86.17 | 49.67 | 51.67 | 25.37 | 89.00 | 53.00 | 89.00 | 87.00 | 83.00 | - | - |
| 10_0.2 | 98.00 | 75.00 | 59.00 | 96.00 | 46.67 | 87.67 | 49.33 | 50.50 | 26.16 | 89.00 | 53.00 | 88.00 | 89.00 | 84.00 | - | - |
| baseline | 97.00 | 67.00 | 61.50 | 90.00 | 48.83 | 87.17 | 48.00 | 51.33 | 26.10 | 88.00 | 58.00 | 90.00 | 90.00 | 85.00 | - | - |
| **Llama-3.2-3B-Instruct** | | | | | | | | | | | | | | | | |
| 20_-0.2 | 96.00 | 35.00 | 98.00 | 63.00 | 50.17 | 85.00 | 48.67 | 48.33 | 13.68 | 80.00 | 54.00 | 91.00 | 87.00 | 82.00 | 19.63 | 35.00 |
| 10_-0.2 | 97.00 | 45.00 | 98.50 | 76.00 | 52.00 | 87.67 | 49.67 | 51.50 | 13.32 | 80.00 | 56.00 | 92.00 | 88.00 | 81.00 | 16.33 | 37.00 |
| 20_0.2 | 95.00 | 62.00 | 98.00 | 79.00 | 49.17 | 84.67 | 54.33 | 52.83 | 20.58 | 83.00 | 59.00 | 90.00 | 88.00 | 83.00 | 18.18 | 34.00 |
| 10_0.2 | 97.00 | 58.00 | 97.75 | 82.00 | 52.00 | 87.17 | 53.67 | 55.50 | 20.77 | 80.00 | 58.00 | 92.00 | 88.00 | 84.00 | 16.88 | 35.00 |
| MI_10 | 97.00 | 58.00 | 98.00 | 78.00 | 51.00 | 85.33 | 52.67 | 51.67 | 21.65 | 81.00 | 57.00 | 90.00 | 88.00 | 83.00 | 20.06 | 31.00 |
| IN_10 | 98.00 | 61.00 | 98.50 | 81.00 | 51.67 | 86.83 | 53.67 | 54.83 | 20.73 | 84.00 | 57.00 | 92.00 | 88.00 | 84.00 | 16.57 | 36.00 |
| baseline | 98.00 | 59.00 | 98.50 | 82.00 | 52.50 | 87.67 | 56.33 | 55.50 | 17.13 | 82.00 | 56.00 | 92.00 | 88.00 | 83.00 | 19.57 | 34.00 |
| **Qwen2.5-7B** | | | | | | | | | | | | | | | | |
| 20_-0.2 | 97.00 | 90.00 | 93.25 | 98.00 | 43.00 | 79.17 | 49.00 | 42.83 | 22.86 | 86.00 | 55.00 | 81.00 | 90.00 | 76.00 | - | - |
| 10_-0.2 | 96.00 | 94.00 | 93.25 | 98.00 | 44.17 | 78.00 | 50.67 | 45.50 | 24.84 | 87.00 | 54.00 | 82.00 | 90.00 | 79.00 | - | - |
| 20_0.2 | 96.00 | 96.00 | 93.50 | 98.00 | 42.50 | 76.83 | 51.00 | 45.83 | 23.25 | 87.00 | 58.00 | 85.00 | 90.00 | 77.00 | - | - |
| 10_0.2 | 97.00 | 94.00 | 93.00 | 98.00 | 42.17 | 77.50 | 51.00 | 45.67 | 23.11 | 87.00 | 58.00 | 86.00 | 90.00 | 79.00 | - | - |
| baseline | 96.00 | 96.00 | 93.00 | 99.00 | 43.00 | 79.00 | 50.33 | 44.50 | 23.15 | 87.00 | 58.00 | 83.00 | 91.00 | 79.00 | - | - |
| **Qwen2.5-7B-Instruct** | | | | | | | | | | | | | | | | |
| 20_-0.2 | 98.00 | 95.00 | 88.50 | 96.00 | 43.17 | 77.67 | 45.67 | 57.00 | 35.88 | 90.00 | 49.00 | 87.00 | 91.00 | 75.00 | 15.45 | 52.00 |
| 10_-0.2 | 98.00 | 97.00 | 89.50 | 97.00 | 42.67 | 78.50 | 45.33 | 59.67 | 35.78 | 90.00 | 49.00 | 86.00 | 91.00 | 76.00 | 14.25 | 53.00 |
| 20_0.2 | 99.00 | 96.00 | 88.75 | 96.00 | 44.17 | 82.00 | 47.33 | 60.50 | 36.75 | 88.00 | 48.00 | 87.00 | 92.00 | 77.00 | 12.90 | 54.00 |
| 10_0.2 | 99.00 | 97.00 | 88.75 | 97.00 | 43.33 | 81.50 | 47.67 | 62.67 | 35.85 | 91.00 | 48.00 | 87.00 | 91.00 | 74.00 | 13.22 | 52.00 |
| MI_10 | 98.00 | 96.00 | 88.75 | 97.00 | 42.00 | 80.50 | 46.33 | 62.50 | 35.87 | 90.00 | 48.00 | 87.00 | 91.00 | 75.00 | 14.16 | 49.00 |
| IN_10 | 98.00 | 97.00 | 89.00 | 97.00 | 41.67 | 80.33 | 46.67 | 62.17 | 35.29 | 90.00 | 47.00 | 85.00 | 90.00 | 75.00 | 13.84 | 52.00 |
| baseline | 97.00 | 97.00 | 89.00 | 97.00 | 42.50 | 80.33 | 47.33 | 60.67 | 36.56 | 91.00 | 45.00 | 86.00 | 90.00 | 75.00 | 14.84 | 49.00 |
| **Ministral-8B-Instruct-2410** | | | | | | | | | | | | | | | | |
| 20_-0.2 | 100.00 | 97.00 | 82.50 | 99.00 | 54.00 | 86.83 | 55.00 | 51.33 | 31.31 | 90.00 | 31.00 | 88.00 | 97.00 | 80.00 | 23.17 | 44.00 |
| 10_-0.2 | 100.00 | 97.00 | 81.25 | 100.00 | 54.33 | 86.83 | 57.00 | 50.50 | 29.74 | 91.00 | 36.00 | 88.00 | 97.00 | 82.00 | 22.34 | 44.00 |
| 20_0.2 | 99.00 | 97.00 | 80.25 | 99.00 | 48.17 | 88.83 | 61.67 | 55.17 | 38.86 | 90.00 | 40.00 | 88.00 | 97.00 | 82.00 | 21.00 | 37.00 |
| 10_0.2 | 99.00 | 97.00 | 80.25 | 99.00 | 49.50 | 89.00 | 61.33 | 55.67 | 36.69 | 93.00 | 39.00 | 88.00 | 97.00 | 81.00 | 21.48 | 36.00 |
| MI_10 | 100.00 | 98.00 | 82.25 | 98.00 | 51.83 | 88.00 | 59.00 | 54.50 | 39.01 | 90.00 | 39.00 | 89.00 | 97.00 | 83.00 | 22.11 | 46.00 |
| IN_10 | 100.00 | 95.00 | 82.25 | 98.00 | 52.00 | 86.33 | 59.00 | 53.83 | 37.44 | 89.00 | 34.00 | 88.00 | 95.00 | 80.00 | 23.20 | 46.00 |
| baseline | 100.00 | 96.00 | 83.00 | 100.00 | 52.17 | 88.50 | 59.00 | 54.67 | 33.68 | 92.00 | 36.00 | 87.00 | 97.00 | 83.00 | 22.12 | 41.00 |

Table 7: Results of HELMET benchmark under 32k context.

| model | Recall | | | | RAG | | | | Re-ranking | | | ICL | | | Long QA | |
| | MK Needle | MK UUID | MV | JSON KV | NQ | TriviaQA | HotpotQA | PopQA | MS MARCO | TREC Coarse | TREC Fine | BANKING77 | CLINC150 | NLU | Infbench QA | Infbench MC |
|---|---|---|---|---|---|---|---|---|---|---|---|---|---|---|---|---|
| **Llama-3.2-3B** | | | | | | | | | | | | | | | | |
| 20_-0.2 | 86.00 | 27.00 | 50.00 | 59.00 | 39.17 | 82.83 | 40.00 | 39.50 | 6.83 | 89.00 | 68.00 | 91.00 | 92.00 | 86.00 | - | - |
| 10_-0.2 | 93.00 | 37.00 | 60.25 | 68.00 | 40.33 | 87.67 | 45.00 | 42.83 | 6.24 | 89.00 | 72.00 | 91.00 | 92.00 | 87.00 | - | - |
| 20_0.2 | 85.00 | 55.00 | 54.25 | 71.00 | 45.33 | 84.17 | 47.33 | 49.00 | 7.08 | 90.00 | 70.00 | 91.00 | 93.00 | 83.00 | - | - |
| 10_0.2 | 87.00 | 51.00 | 54.75 | 70.00 | 44.67 | 83.83 | 45.33 | 49.50 | 9.27 | 90.00 | 69.00 | 92.00 | 92.00 | 84.00 | - | - |
| baseline | 88.00 | 46.00 | 58.00 | 70.00 | 45.33 | 85.50 | 42.33 | 46.17 | 7.29 | 90.00 | 72.00 | 92.00 | 92.00 | 85.00 | - | - |
| **Llama-3.2-3B-Instruct** | | | | | | | | | | | | | | | | |
| 20_-0.2 | 78.00 | 7.00 | 97.50 | 43.00 | 48.50 | 83.83 | 48.00 | 46.67 | 3.77 | 83.00 | 70.00 | 91.00 | 91.00 | 84.00 | 19.27 | 38.00 |
| 10_-0.2 | 82.00 | 13.00 | 99.25 | 50.00 | 48.83 | 86.00 | 50.67 | 47.83 | 2.44 | 86.00 | 68.00 | 93.00 | 93.00 | 85.00 | 18.76 | 42.00 |
| 20_0.2 | 82.00 | 22.00 | 99.25 | 60.00 | 48.00 | 85.83 | 52.33 | 50.67 | 2.72 | 85.00 | 64.00 | 92.00 | 90.00 | 86.00 | 18.45 | 38.00 |
| 10_0.2 | 82.00 | 20.00 | 99.00 | 56.00 | 49.83 | 86.83 | 51.67 | 50.83 | 3.10 | 85.00 | 67.00 | 92.00 | 91.00 | 86.00 | 19.60 | 34.00 |
| MI_10 | 81.00 | 19.00 | 98.00 | 60.00 | 51.00 | 86.17 | 49.33 | 49.50 | 3.13 | 85.00 | 68.00 | 92.00 | 90.00 | 86.00 | 18.59 | 32.00 |
| IN_10 | 82.00 | 23.00 | 99.50 | 60.00 | 49.67 | 87.00 | 54.33 | 49.67 | 4.16 | 85.00 | 69.00 | 91.00 | 91.00 | 84.00 | 19.14 | 39.00 |
| baseline | 82.00 | 19.00 | 99.50 | 56.00 | 48.50 | 86.83 | 53.67 | 50.83 | 3.77 | 85.00 | 70.00 | 94.00 | 92.00 | 84.00 | 19.27 | 43.00 |
| **Qwen2.5-7B** | | | | | | | | | | | | | | | | |
| 20_-0.2 | 31.00 | 29.00 | 84.25 | 24.00 | 29.33 | 66.17 | 40.67 | 31.67 | 1.30 | 85.00 | 53.00 | 84.00 | 88.00 | 77.00 | - | - |
| 10_-0.2 | 39.00 | 28.00 | 85.00 | 28.00 | 31.67 | 67.67 | 41.67 | 32.67 | 2.64 | 86.00 | 53.00 | 82.00 | 88.00 | 78.00 | - | - |
| 20_0.2 | 47.00 | 26.00 | 85.25 | 28.00 | 30.17 | 67.50 | 41.67 | 33.00 | 1.19 | 87.00 | 53.00 | 80.00 | 88.00 | 80.00 | - | - |
| 10_0.2 | 43.00 | 28.00 | 86.25 | 29.00 | 29.83 | 67.67 | 43.33 | 33.67 | 1.07 | 86.00 | 54.00 | 84.00 | 90.00 | 79.00 | - | - |
| baseline | 41.00 | 28.00 | 83.75 | 28.00 | 31.33 | 68.17 | 43.33 | 33.67 | 1.88 | 86.00 | 53.00 | 83.00 | 89.00 | 79.00 | - | - |
| **Qwen2.5-7B-Instruct** | | | | | | | | | | | | | | | | |
| 20_-0.2 | 40.00 | 32.00 | 83.25 | 30.00 | 29.83 | 67.17 | 39.67 | 39.17 | 11.92 | 85.00 | 53.00 | 86.00 | 91.00 | 79.00 | 7.15 | 37.00 |
| 10_-0.2 | 44.00 | 29.00 | 82.50 | 30.00 | 31.50 | 67.83 | 38.33 | 40.00 | 12.21 | 86.00 | 51.00 | 87.00 | 91.00 | 78.00 | 7.33 | 36.00 |
| 20_0.2 | 63.00 | 28.00 | 81.50 | 33.00 | 36.33 | 69.50 | 39.67 | 42.00 | 10.28 | 87.00 | 51.00 | 87.00 | 90.00 | 76.00 | 6.89 | 39.00 |
| 10_0.2 | 56.00 | 28.00 | 81.00 | 32.00 | 35.50 | 69.50 | 42.67 | 43.50 | 11.66 | 87.00 | 50.00 | 86.00 | 91.00 | 77.00 | 7.88 | 40.00 |
| MI_10 | 44.00 | 30.00 | 82.00 | 30.00 | 32.67 | 68.67 | 39.67 | 42.33 | 12.32 | 87.00 | 50.00 | 87.00 | 92.00 | 79.00 | 7.31 | 37.00 |
| IN_10 | 47.00 | 31.00 | 81.25 | 32.00 | 35.17 | 69.83 | 42.67 | 41.67 | 11.84 | 86.00 | 49.00 | 86.00 | 91.00 | 77.00 | 6.51 | 36.00 |
| baseline | 48.00 | 31.00 | 81.50 | 31.00 | 33.83 | 68.17 | 41.67 | 42.50 | 11.88 | 87.00 | 50.00 | 88.00 | 91.00 | 77.00 | 6.86 | 38.00 |
| **Ministral-8B-Instruct-2410** | | | | | | | | | | | | | | | | |
| 20_-0.2 | 45.00 | 9.00 | 55.25 | 13.00 | 34.50 | 72.00 | 40.00 | 38.17 | 0.00 | 87.00 | 58.00 | 91.00 | 86.00 | 81.00 | 16.82 | 26.00 |
| 10_-0.2 | 44.00 | 8.00 | 55.25 | 13.00 | 34.00 | 72.33 | 40.00 | 37.83 | 0.00 | 86.00 | 59.00 | 89.00 | 86.00 | 80.00 | 16.24 | 25.00 |
| 20_0.2 | 42.00 | 10.00 | 56.50 | 15.00 | 33.17 | 74.67 | 41.67 | 39.00 | 0.00 | 88.00 | 65.00 | 87.00 | 91.00 | 81.00 | 15.95 | 24.00 |
| 10_0.2 | 43.00 | 9.00 | 56.75 | 16.00 | 34.50 | 72.50 | 42.33 | 37.83 | 0.00 | 88.00 | 65.00 | 87.00 | 91.00 | 83.00 | 15.99 | 23.00 |
| MI_10 | 44.00 | 7.00 | 55.25 | 14.00 | 36.00 | 72.83 | 41.67 | 36.67 | 0.00 | 86.00 | 60.00 | 87.00 | 88.00 | 81.00 | 16.71 | 25.00 |
| IN_10 | 42.00 | 6.00 | 54.75 | 11.00 | 36.83 | 73.00 | 41.33 | 37.17 | 0.00 | 86.00 | 58.00 | 88.00 | 91.00 | 80.00 | 16.60 | 25.00 |
| baseline | 44.00 | 7.00 | 56.50 | 15.00 | 35.67 | 73.17 | 42.00 | 37.83 | 0.00 | 87.00 | 63.00 | 87.00 | 89.00 | 81.00 | 16.79 | 26.00 |

Table 8: Results of HELMET benchmark under 64k context.

| model | Recall | | | | RAG | | | | Re-ranking | | | ICL | | | Long QA | |
|---|---|---|---|---|---|---|---|---|---|---|---|---|---|---|---|---|
| | MK Needle | MK UUID | MV | JSON KV | NQ | TriviaQA | HotpotQA | PopQA | MS MARCO | TREC Coarse | TREC Fine | BANKING77 | CLINC150 | NLU | Infbench QA | Infbench MC |
| **Llama-3.2-3B** | | | | | | | | | | | | | | | | |
| 20_-0.2 | 72.00 | 13.00 | 47.00 | 57.00 | 35.83 | 77.17 | 37.33 | 40.67 | 4.74 | 89.00 | 77.00 | 92.00 | 92.00 | 88.00 | - | - |
| 10_-0.2 | 76.00 | 20.00 | 51.75 | 59.00 | 35.33 | 80.83 | 41.67 | 42.67 | 4.62 | 90.00 | 81.00 | 92.00 | 91.00 | 87.00 | - | - |
| 20_0.2 | 84.00 | 23.00 | 61.50 | 60.00 | 38.33 | 81.33 | 43.00 | 42.00 | 6.78 | 89.00 | 76.00 | 92.00 | 93.00 | 89.00 | - | - |
| 10_0.2 | 87.00 | 25.00 | 53.75 | 62.00 | 37.00 | 81.67 | 44.00 | 43.00 | 6.56 | 89.00 | 76.00 | 93.00 | 93.00 | 90.00 | - | - |
| baseline | 86.00 | 26.00 | 58.00 | 64.00 | 37.17 | 81.83 | 43.33 | 42.33 | 5.53 | 90.00 | 81.00 | 93.00 | 93.00 | 89.00 | - | - |
| **Llama-3.2-3B-Instruct** | | | | | | | | | | | | | | | | |
| 20_-0.2 | 60.00 | 5.00 | 89.50 | 32.00 | 44.17 | 77.17 | 44.00 | 39.50 | 0.32 | 83.00 | 67.00 | 90.00 | 93.00 | 90.00 | 20.50 | 39.00 |
| 10_-0.2 | 68.00 | 7.00 | 95.00 | 33.00 | 43.33 | 81.83 | 46.33 | 43.17 | 0.05 | 83.00 | 67.00 | 92.00 | 93.00 | 91.00 | 19.39 | 42.00 |
| 20_0.2 | 75.00 | 6.00 | 96.25 | 44.00 | 43.33 | 81.33 | 49.33 | 45.33 | 0.00 | 84.00 | 66.00 | 91.00 | 92.00 | 91.00 | 18.29 | 31.00 |
| 10_0.2 | 75.00 | 7.00 | 96.00 | 42.00 | 44.00 | 83.33 | 51.00 | 46.00 | 0.26 | 84.00 | 67.00 | 92.00 | 92.00 | 90.00 | 19.37 | 28.00 |
| MI_10 | 78.00 | 6.00 | 93.00 | 44.00 | 42.00 | 79.50 | 48.33 | 43.17 | 0.00 | 85.00 | 66.00 | 92.00 | 91.00 | 91.00 | 19.11 | 27.00 |
| IN_10 | 75.00 | 8.00 | 98.25 | 41.00 | 45.33 | 82.67 | 51.67 | 44.50 | 0.20 | 84.00 | 69.00 | 93.00 | 92.00 | 91.00 | 19.51 | 42.00 |
| baseline | 73.00 | 5.00 | 97.50 | 39.00 | 45.83 | 85.00 | 50.00 | 45.50 | 0.74 | 83.00 | 70.00 | 91.00 | 94.00 | 92.00 | 19.46 | 37.00 |
| **Qwen2.5-7B** | | | | | | | | | | | | | | | | |
| 20_-0.2 | 8.00 | 5.00 | 30.50 | 12.00 | 18.50 | 42.00 | 29.00 | 24.83 | 0.24 | 76.00 | 46.00 | 59.00 | 82.00 | 76.00 | - | - |
| 10_-0.2 | 9.00 | 5.00 | 30.75 | 13.00 | 16.33 | 42.83 | 25.67 | 24.83 | 0.12 | 77.00 | 43.00 | 61.00 | 84.00 | 74.00 | - | - |
| 20_0.2 | 9.00 | 0.00 | 30.75 | 11.00 | 15.33 | 43.67 | 23.67 | 24.67 | 0.31 | 74.00 | 46.00 | 58.00 | 83.00 | 77.00 | - | - |
| 10_0.2 | 8.00 | 1.00 | 31.00 | 13.00 | 15.00 | 43.50 | 26.33 | 25.00 | 0.00 | 75.00 | 48.00 | 60.00 | 81.00 | 74.00 | - | - |
| baseline | 9.00 | 2.00 | 30.75 | 13.00 | 15.33 | 42.33 | 25.67 | 26.33 | 0.00 | 77.00 | 45.00 | 62.00 | 82.00 | 74.00 | - | - |
| **Qwen2.5-7B-Instruct** | | | | | | | | | | | | | | | | |
| 20_-0.2 | 5.00 | 0.00 | 24.75 | 8.00 | 21.17 | 52.17 | 22.00 | 24.83 | 1.25 | 79.00 | 40.00 | 68.00 | 81.00 | 79.00 | 5.52 | 33.00 |
| 10_-0.2 | 8.00 | 2.00 | 26.50 | 10.00 | 21.00 | 54.50 | 20.00 | 26.50 | 1.34 | 81.00 | 42.00 | 64.00 | 83.00 | 80.00 | 5.39 | 35.00 |
| 20_0.2 | 8.00 | 1.00 | 27.50 | 9.00 | 19.50 | 54.17 | 22.33 | 28.50 | 1.92 | 76.00 | 46.00 | 65.00 | 74.00 | 78.00 | 5.35 | 40.00 |
| 10_0.2 | 8.00 | 1.00 | 28.25 | 9.00 | 21.67 | 54.33 | 22.67 | 28.33 | 2.25 | 77.00 | 43.00 | 65.00 | 80.00 | 78.00 | 5.11 | 38.00 |
| MI_10 | 8.00 | 0.00 | 27.00 | 9.00 | 21.33 | 56.17 | 21.33 | 26.50 | 1.43 | 79.00 | 44.00 | 64.00 | 81.00 | 78.00 | 5.34 | 35.00 |
| IN_10 | 9.00 | 1.00 | 26.50 | 8.00 | 22.33 | 53.50 | 21.67 | 28.50 | 1.74 | 79.00 | 43.00 | 63.00 | 81.00 | 79.00 | 5.15 | 36.00 |
| baseline | 8.00 | 2.00 | 26.50 | 10.00 | 20.83 | 53.67 | 22.00 | 27.67 | 1.82 | 79.00 | 41.00 | 65.00 | 82.00 | 78.00 | 5.10 | 37.00 |
| **Ministral-8B-Instruct-2410** | | | | | | | | | | | | | | | | |
| 20_-0.2 | 15.00 | 4.00 | 22.75 | 10.00 | 19.67 | 56.33 | 32.00 | 30.67 | 0.00 | 86.00 | 67.00 | 81.00 | 93.00 | 77.00 | 12.51 | 29.00 |
| 10_-0.2 | 15.00 | 4.00 | 22.50 | 11.00 | 22.00 | 56.50 | 32.33 | 31.67 | 0.00 | 87.00 | 68.00 | 82.00 | 93.00 | 76.00 | 12.55 | 29.00 |
| 20_0.2 | 13.00 | 2.00 | 23.00 | 9.00 | 21.67 | 57.33 | 34.00 | 29.50 | 0.00 | 89.00 | 68.00 | 84.00 | 94.00 | 77.00 | 12.88 | 29.00 |
| 10_0.2 | 13.00 | 2.00 | 23.00 | 9.00 | 21.17 | 57.67 | 32.33 | 30.50 | 0.00 | 87.00 | 67.00 | 83.00 | 94.00 | 80.00 | 13.63 | 30.00 |
| MI_10 | 16.00 | 4.00 | 23.00 | 9.00 | 21.17 | 57.17 | 33.33 | 29.33 | 0.00 | 86.00 | 65.00 | 79.00 | 94.00 | 78.00 | 12.10 | 32.00 |
| IN_10 | 16.00 | 3.00 | 23.25 | 9.00 | 21.50 | 56.83 | 32.00 | 29.33 | 0.00 | 85.00 | 69.00 | 80.00 | 92.00 | 78.00 | 12.55 | 30.00 |
| baseline | 14.00 | 4.00 | 23.00 | 10.00 | 20.17 | 57.50 | 35.00 | 30.17 | 0.00 | 87.00 | 69.00 | 82.00 | 94.00 | 79.00 | 12.40 | 31.00 |

Table 9: Results of HELMET benchmark under 128k context.

| Model | Recall | RAG | Re-ranking | ICL | Long QA | Overall Average |
|---|---|---|---|---|---|---|
| **Llama-3.2-3B** | | | | | | |
| 20_-0.2 | 82.50 | 58.00 | 38.15 | 60.20 | - | 59.71 |
| 10_-0.2 | 88.62 | 59.42 | 42.31 | 61.20 | - | 62.89 |
| 20_0.2 | 89.38 | 61.42 | 46.21 | 65.80 | - | 65.70 |
| 10_0.2 | 90.81 | 62.12 | 42.41 | 66.00 | - | 65.34 |
| baseline | 90.62 | 61.46 | 43.52 | 64.40 | - | 65.00 |
| **Llama-3.2-3B-Instruct** | | | | | | |
| 20_-0.2 | 94.25 | 61.25 | 35.19 | 66.00 | 25.23 | 56.38 |
| 10_-0.2 | 98.25 | 62.79 | 36.71 | 65.20 | 26.49 | 57.89 |
| 20_0.2 | 98.44 | 65.96 | 48.37 | 65.00 | 22.81 | 60.11 |
| 10_0.2 | 98.25 | 66.62 | 46.90 | 66.20 | 20.44 | 59.68 |
| MI_10 | 97.75 | 66.25 | 46.37 | 66.20 | 20.95 | 59.51 |
| IN_10 | 98.44 | 65.62 | 47.15 | 64.80 | 21.91 | 59.58 |
| baseline | 99.25 | 65.33 | 41.68 | 65.40 | 23.24 | 58.98 |
| **Qwen2.5-7B** | | | | | | |
| 20_-0.2 | 97.75 | 61.25 | 48.67 | 68.20 | - | 68.97 |
| 10_-0.2 | 98.38 | 61.92 | 48.55 | 68.80 | - | 69.41 |
| 20_0.2 | 97.81 | 62.71 | 44.94 | 67.60 | - | 68.26 |
| 10_0.2 | 97.81 | 62.92 | 46.52 | 68.80 | - | 69.01 |
| baseline | 98.50 | 62.79 | 47.44 | 68.60 | - | 69.33 |
| **Qwen2.5-7B-Instruct** | | | | | | |
| 20_-0.2 | 98.50 | 62.08 | 58.53 | 71.00 | 32.09 | 64.44 |
| 10_-0.2 | 98.62 | 62.58 | 56.51 | 71.00 | 32.22 | 64.19 |
| 20_0.2 | 98.81 | 65.29 | 56.25 | 69.40 | 32.60 | 64.47 |
| 10_0.2 | 98.81 | 65.12 | 56.98 | 70.20 | 33.27 | 64.88 |
| MI_10 | 99.12 | 64.79 | 57.33 | 70.80 | 32.45 | 64.90 |
| IN_10 | 99.12 | 64.71 | 57.76 | 70.60 | 32.80 | 65.00 |
| baseline | 98.94 | 64.25 | 57.54 | 70.80 | 32.06 | 64.72 |
| **Ministral-8B-Instruct-2410** | | | | | | |
| 20_-0.2 | 98.44 | 66.04 | 65.00 | 71.60 | 31.15 | 66.45 |
| 10_-0.2 | 98.69 | 65.46 | 64.95 | 71.80 | 31.94 | 66.57 |
| 20_0.2 | 98.88 | 67.21 | 66.88 | 72.00 | 29.52 | 66.90 |
| 10_0.2 | 98.81 | 66.92 | 65.80 | 71.60 | 28.19 | 66.26 |
| MI_10 | 98.81 | 66.92 | 67.59 | 72.00 | 30.44 | 67.15 |
| IN_10 | 98.88 | 67.21 | 68.42 | 71.40 | 30.68 | 67.32 |
| baseline | 99.00 | 66.92 | 66.32 | 71.80 | 32.05 | 67.22 |

Table 10: Category average results of HELMET benchmark under 8k context.

| Model | Recall | RAG | Re-ranking | ICL | Long QA | Overall Average |
|---|---|---|---|---|---|---|
| **Llama-3.2-3B** | | | | | | |
| 20_-0.2 | 55.50 | 50.38 | 6.83 | 85.20 | - | 49.48 |
| 10_-0.2 | 64.56 | 53.96 | 6.24 | 86.20 | - | 52.74 |
| 20_0.2 | 66.31 | 56.46 | 7.08 | 85.40 | - | 53.81 |
| 10_0.2 | 65.69 | 55.83 | 9.27 | 85.40 | - | 54.05 |
| baseline | 65.50 | 54.83 | 7.29 | 86.20 | - | 53.46 |
| **Llama-3.2-3B-Instruct** | | | | | | |
| 20_-0.2 | 56.38 | 56.75 | 3.77 | 83.80 | 28.64 | 45.87 |
| 10_-0.2 | 61.06 | 58.33 | 2.44 | 85.00 | 30.38 | 47.44 |
| 20_0.2 | 65.81 | 59.21 | 2.72 | 83.40 | 28.23 | 47.87 |
| 10_0.2 | 64.25 | 59.79 | 3.10 | 84.20 | 26.80 | 47.63 |
| MI_10 | 64.50 | 59.00 | 3.13 | 84.20 | 25.29 | 47.23 |
| IN_10 | 66.12 | 60.17 | 4.16 | 84.00 | 29.07 | 48.70 |
| baseline | 64.12 | 59.96 | 3.77 | 85.00 | 31.13 | 48.80 |
| **Qwen2.5-7B** | | | | | | |
| 20_-0.2 | 42.06 | 41.96 | 1.30 | 77.40 | - | 40.68 |
| 10_-0.2 | 45.00 | 43.42 | 2.64 | 77.40 | - | 42.11 |
| 20_0.2 | 46.56 | 43.08 | 1.19 | 77.60 | - | 42.11 |
| 10_0.2 | 46.56 | 43.62 | 1.07 | 78.80 | - | 42.51 |
| baseline | 45.19 | 44.12 | 1.88 | 78.00 | - | 42.30 |
| **Qwen2.5-7B-Instruct** | | | | | | |
| 20_-0.2 | 46.31 | 43.96 | 11.92 | 78.80 | 22.07 | 40.61 |
| 10_-0.2 | 46.38 | 44.42 | 12.21 | 78.60 | 21.66 | 40.65 |
| 20_0.2 | 51.38 | 46.88 | 10.28 | 78.20 | 22.95 | 41.94 |
| 10_0.2 | 49.25 | 47.79 | 11.66 | 78.20 | 23.94 | 42.17 |
| MI_10 | 46.50 | 45.83 | 12.32 | 79.00 | 22.16 | 41.16 |
| IN_10 | 47.81 | 47.33 | 11.84 | 77.80 | 21.25 | 41.21 |
| baseline | 47.88 | 46.54 | 11.88 | 78.60 | 22.43 | 41.46 |
| **Ministral-8B-Instruct-2410** | | | | | | |
| 20_-0.2 | 30.56 | 46.17 | 0.00 | 80.60 | 21.41 | 35.75 |
| 10_-0.2 | 30.06 | 46.04 | 0.00 | 80.00 | 20.62 | 35.34 |
| 20_0.2 | 30.88 | 47.12 | 0.00 | 81.80 | 19.98 | 35.96 |
| 10_0.2 | 31.19 | 46.79 | 0.00 | 82.80 | 19.49 | 36.05 |
| MI_10 | 30.06 | 46.79 | 0.00 | 80.40 | 20.86 | 35.62 |
| IN_10 | 28.44 | 47.08 | 0.00 | 80.60 | 20.80 | 35.38 |
| baseline | 30.62 | 47.17 | 0.00 | 81.40 | 21.40 | 36.12 |

Table 11: Category average results of HELMET benchmark under 64k context.

| Model | Recall | RAG | Re-ranking | ICL | Long QA | Overall Average |
|---|---|---|---|---|---|---|
| **Llama-3.2-3B** | | | | | | |
| 20_-0.2 | 47.25 | 47.75 | 4.74 | 87.60 | - | 46.83 |
| 10_-0.2 | 51.69 | 50.12 | 4.62 | 88.20 | - | 48.66 |
| 20_0.2 | 57.12 | 51.17 | 6.78 | 87.80 | - | 50.72 |
| 10_0.2 | 56.94 | 51.42 | 6.56 | 88.20 | - | 50.78 |
| baseline | 58.50 | 51.17 | 5.53 | 89.20 | - | 51.10 |
| **Llama-3.2-3B-Instruct** | | | | | | |
| 20_-0.2 | 46.62 | 51.21 | 0.32 | 84.60 | 29.75 | 42.50 |
| 10_-0.2 | 50.75 | 53.67 | 0.05 | 85.20 | 30.70 | 44.07 |
| 20_0.2 | 55.31 | 54.83 | 0.00 | 84.80 | 24.64 | 43.92 |
| 10_0.2 | 55.00 | 56.08 | 0.26 | 85.00 | 23.69 | 44.01 |
| MI_10 | 55.25 | 53.25 | 0.00 | 85.00 | 23.06 | 43.31 |
| IN_10 | 55.56 | 56.04 | 0.20 | 85.80 | 30.76 | 45.67 |
| baseline | 53.62 | 56.58 | 0.74 | 86.00 | 28.23 | 45.04 |
| **Qwen2.5-7B** | | | | | | |
| 20_-0.2 | 13.88 | 28.58 | 0.24 | 67.80 | - | 27.62 |
| 10_-0.2 | 14.44 | 27.42 | 0.12 | 67.80 | - | 27.44 |
| 20_0.2 | 12.69 | 26.83 | 0.31 | 67.60 | - | 26.86 |
| 10_0.2 | 13.25 | 27.46 | 0.00 | 67.60 | - | 27.08 |
| baseline | 13.69 | 27.42 | 0.00 | 68.00 | - | 27.28 |
| **Qwen2.5-7B-Instruct** | | | | | | |
| 20_-0.2 | 9.44 | 30.04 | 1.25 | 69.40 | 19.26 | 25.88 |
| 10_-0.2 | 11.62 | 30.50 | 1.34 | 70.00 | 20.19 | 26.73 |
| 20_0.2 | 11.38 | 31.12 | 1.92 | 67.80 | 22.68 | 26.98 |
| 10_0.2 | 11.56 | 31.75 | 2.25 | 68.60 | 21.56 | 27.14 |
| MI_10 | 11.00 | 31.33 | 1.43 | 69.20 | 20.17 | 26.63 |
| IN_10 | 11.12 | 31.50 | 1.74 | 69.00 | 20.57 | 26.79 |
| baseline | 11.62 | 31.04 | 1.82 | 69.00 | 21.05 | 26.91 |
| **Ministral-8B-Instruct-2410** | | | | | | |
| 20_-0.2 | 12.94 | 34.67 | 0.00 | 80.80 | 20.76 | 29.83 |
| 10_-0.2 | 13.12 | 35.62 | 0.00 | 81.20 | 20.77 | 30.14 |
| 20_0.2 | 11.75 | 35.62 | 0.00 | 82.40 | 20.94 | 30.14 |
| 10_0.2 | 11.75 | 35.42 | 0.00 | 82.20 | 21.81 | 30.24 |
| MI_10 | 13.00 | 35.25 | 0.00 | 80.40 | 22.05 | 30.14 |
| IN_10 | 12.81 | 34.92 | 0.00 | 80.80 | 21.28 | 29.96 |
| baseline | 12.75 | 35.71 | 0.00 | 82.20 | 21.70 | 30.47 |

Table 12: Category average results of HELMET benchmark under 128k context.

| model | MK Needle | MK UUID | MV | JSON KV | NQ | TriviaQA | HotpotQA | PopQA | MS MARCO | TREC Coarse | TREC Fine | BANKING77 | CLINC150 | NLU | Infbench QA | Infbench MC | STD |
|---|---|---|---|---|---|---|---|---|---|---|---|---|---|---|---|---|---|
| Llama-3.2-3B | 21.91 | 25.35 | 23.94 | 23.19 | 20.44 | 21.88 | 20.91 | 23.70 | 23.74 | 29.52 | 30.02 | 9.78 | 26.57 | 23.73 | - | - | 4.81 |
| Llama-3.2-3B-Instruct | 31.85 | 29.34 | 24.42 | 34.02 | 22.53 | 26.78 | 24.81 | 25.90 | 30.04 | 53.52 | 35.39 | 26.45 | 31.76 | 31.67 | 18.61 | 28.49 | 7.71 |
| Qwen2.5-7B | 16.80 | 6.39 | 8.35 | 6.61 | 9.25 | 10.63 | 8.82 | 10.03 | 5.35 | 31.29 | 20.61 | 3.40 | 11.72 | 10.63 | - | - | 7.24 |
| Qwen2.5-7B-Instruct | 20.07 | 18.41 | 10.42 | 6.87 | 6.97 | 8.89 | 9.10 | 10.69 | 4.18 | 33.32 | 23.57 | 3.58 | 9.67 | 7.86 | 0.17 | 0.28 | 8.87 |
| Ministral-8B-Instruct-2410 | 22.45 | 15.48 | 14.76 | 14.56 | 11.59 | 16.52 | 15.89 | 20.63 | 12.32 | 26.98 | 12.92 | 9.70 | 16.21 | 14.71 | 17.16 | 20.32 | 4.39 |

Table 13: Sink contextual scores (%) and its standard deviation (STD) under 8k contexts (average of top-5 contextual heads).

| model | MK Needle | MK UUID | MV | JSON KV | NQ | TriviaQA | HotpotQA | PopQA | MS MARCO | TREC Coarse | TREC Fine | BANKING77 | CLINC150 | NLU | Infbench QA | Infbench MC | STD |
|---|---|---|---|---|---|---|---|---|---|---|---|---|---|---|---|---|---|
| Llama-3.2-3B | 21.60 | 23.61 | 26.68 | 22.63 | 19.63 | 21.51 | 19.82 | 21.83 | 22.43 | 29.28 | 23.04 | 7.78 | 19.70 | 22.22 | - | - | 7.21 |
| Llama-3.2-3B-Instruct | 32.53 | 23.09 | 23.01 | 29.37 | 22.28 | 25.65 | 24.15 | 24.00 | 32.52 | 44.59 | 36.98 | 22.30 | 24.11 | 25.65 | 17.69 | 31.37 | 9.33 |
| Qwen2.5-7B | 13.61 | 5.25 | 6.44 | 6.43 | 8.68 | 9.71 | 7.74 | 9.15 | 4.53 | 30.56 | 6.50 | 2.80 | 4.49 | 7.73 | - | - | 6.94 |
| Qwen.Qwen2.5-7B-Instruct | 9.45 | 5.00 | 8.04 | 4.89 | 7.55 | 10.15 | 8.57 | 9.23 | 5.98 | 19.65 | 6.96 | 2.45 | 4.81 | 6.96 | 0.25 | 0.43 | 4.70 |
| Ministral-8B-Instruct-2410 | 19.87 | 12.69 | 12.41 | - | - | 13.26 | 13.02 | 16.48 | 10.44 | 16.29 | 12.64 | 6.81 | 11.28 | 13.35 | 11.02 | 15.21 | 4.55 |

Table 14: Sink contextual scores (%) and its standard deviation (STD) under 16k contexts (average of top-5 contextual heads).

