# OpenReview forum: "Focus Directions Make Your Language Models Pay More Attention to Relevant Contexts"
_ICLR.cc/2026/Conference — ICLR 2026 Conference Withdrawn Submission_

### Official Review · Reviewer_X4Ze · 2025-10-29

**Soundness:** 2
**Presentation:** 3
**Contribution:** 2
**Rating:** 4
**Confidence:** 3

**Summary:**

This paper finds that the distraction happens when these Contextual Heads fail to pay enough attention to the relevant context. It then identifies Focus Directions, which are specific directional vectors located in the key and query activations of these heads, that directly control the attention mechanism. They also introduced an automated magnitude control method to determine the optimal strength for applying the Focus Directions.

**Strengths:**

1. The central idea of introducing Automated Magnitude Control to dynamically adjust the strength of the intervention (Focus Directions) is novel and addresses a practical challenge in applying direct attention-head steering, making the method more robust and less reliant on manual hyperparameter tuning.
2. The authors have provided a large volume of detailed experiments across several LLMs and context lengths. The presentation of the methodology and results is clear, making the paper relatively easy to follow

**Weaknesses:**

1. The experiments predominantly focus on the simple one-hop Needle-in-a-Haystack (NIAH) task. It is a major concern whether the proposed methodology can effectively scale to more challenging, real-world long-context tasks, such as multi-hop reasoning or complex question answering, where the relevant context is distributed and requires multiple retrieval steps.

2. The method's performance improvement appears to be inconsistent and sometimes non-existent across various context lengths. This lack of a stable, monotonic improvement raises serious concerns regarding the overall robustness and reliability of the proposed approach under diverse operational constraints.

**Questions:**

please refer to weaknesses.

---

> ### Author Response · Authors · 2025-12-01
>
> W1: results in multiple retrieval steps
>
> Tasks with multiple retrieval steps are actually covered in HELMET benchmarks (e.g., HotpotQA). Focus directions show reasonable improvement in such a case. For example, in Table 5, we show the result of Llama-3.2-3B improved from 52.67 (baseline) to 56.0.
>
>
>
> W2: lack of a stable, monotonic improvement
>
> Thanks for the comment. First, we clarify that our obtained focus directions have strong effectiveness in-domain (Section 3) and some transferability out-of-domain (Section 4). While the out-of-domain improvement is not as strong as in-domain, we consider this a basic property of focus directions. This work demonstrates the existence and effectiveness of focus direction under in-domain settings. Future work could further explore multiple focus directions for different domains and switching between different directions.

---

### Official Review · Reviewer_QhmA · 2025-10-30

**Soundness:** 3
**Presentation:** 2
**Contribution:** 2
**Rating:** 4
**Confidence:** 4

**Summary:**

The paper tackles long-context getting distracted by finding the few attention heads that naturally focus on the right spans and then gently steering them at inference time. The authors learn simple focus directions in those heads so the model pays less attention to sink tokens and more to likely relevant text, without finetuning or editing the input. Across multi-document QA and several long-context benchmarks, this yields consistent, interpretable gains. Overall, it’s a lightweight, diagnostics-friendly way to help LMs look in the right place.

**Strengths:**

1. The work gives a simple, inference-time way to nudge a few attention heads, so it can get gains without finetuning or rewriting the input.

2. On multi-document QA and several HELMET tasks, it improves accuracy under long contexts across different models and context lengths.

3. The head scoring and attention evidence are easy to understand.

**Weaknesses:**

1. The method requires labeled relevant span supervision (e.g., the 20-doc, 1-relevant setup) to identify contextual heads and learn focus directions, so transfer to domains without such supervision is uncertain.

2. Performance depends strongly on the number of intervened heads and the intervention magnitude, and small changes can flip gains into regressions, implying per-model and per-task tuning is necessary.

3. Deployment becomes difficult for closed-weight models because the approach needs access to internal activations to intervene at inference.

4. The learned directions likely favor semantic overlap with the query, which can reduce robustness on paraphrases or counterfactual spans where evidence is indirect.

5. Improvements are uneven across tasks and models, leaving unclear guidance on when to enable the method, which heads to pick, and what intervention strength to use.

**Questions:**

1. How would the method work in domains without labeled relevant spans?

2. Does the approach bias toward lexical overlap with the query, and can you report robustness on paraphrase and counterfactual setups where evidence is indirect?

3. Given the uneven gains across tasks and models, can you provide a simple decision rule or usage guideline (when to enable, which heads, what α) validated across datasets?

---

> ### Author Response · Authors · 2025-12-01
>
> W1: require relevant span supervision, uncertain transferability
>
> While we agree that the current method requires labeled relevant spans to obtain focus directions, obtaining focus directions without labeled relevant spans could be further explored with future work (e.g., via synthetic data). In addition, this paper aims to reveal the existence and basic properties of the focus directions, in which we show that focus directions are able to transfer to unseen tasks to a certain extent (Section 4).
>
>
> W2: performance depends on the number of heads and the magnitude
>
> Thanks for pointing that out. We consider this a property of the focus directions. The focus directions control the amount of attention activation from the attention sink to the contexts.
> Such amount of attention activation may not be optimal for some pre-trained or fine-tuned LLMs under certain tasks and thus result in not optimal downstream performance. In such cases, focus directions help to align LLMs to the optimal attention activation.
>
> In addition, we introduce an automatic magnitude control method that eliminates the need to tune the magnitude.
>
> W3: difficult development for close weight models
>
> Thanks for pointing that out. The main goal of this paper is to study focus directions as an internal mechanism of the LLMs. Close weight model developers could benefit from our findings to develop better models.
>
>
>
> W4: learned directions likely favor semantic overlap with the query
>
> We do not think focus directions favor the semantic overlap. To show this, we paraphrase the question in our multi-document QA data (test set) and evaluate using the Llama-3.2-3B-Instruct model. Results are 59.6% (baseline), 64.4% (10_0.2), 64.9% (20_0.2), similar to the original result.
>
>
> W5: unclear guidance on hyperparameters
>
> To address the challenge of hyperparameter tuning, we introduced an automatic magnitude control method in Section 3.5. Results show that automatic magnitude control improves performance without the need for manually setting magnitude. For example, in Table 3, Llama-3.2-3B-Instruct improved from 59.4% to 66.4% with automatic magnitude control.
>
>
> Q1: How would the method work in domains without labeled relevant spans
>
> Our method contains two stages: obtaining focus directions (training) and applying focus directions (inference).
>
> Obtaining focus directions: labeled relevant spans are required at this stage. We obtain the focus direction by enlarging the attention to the relevant contexts.
>
> Applying focus directions: labeled relevant spans are not needed at this stage. The learned focus directions are already able to make the LLM pay more attention to the relevant contexts. This can be generalized to the unseen data.
>
>
> Q2: bias toward lexical overlap
>
> See W4.
>
>
> Q3: decision rule/usage guideline
>
> See W5.

---

### Official Review · Reviewer_xHAX · 2025-10-30

**Soundness:** 2
**Presentation:** 3
**Contribution:** 2
**Rating:** 4
**Confidence:** 3

**Summary:**

This paper investigate the context distraction issue in long-context LLMs, where the relevent parts for QA burried in a large portion of irrelevent parts of the input. The authors first identify contextal heads, which are a small subsets of attention heads that are senitive in focusing on relevant spans of input. Then they introduce applying focus direction on those attention head by adding a vectors to the key and query representations, this aims to steer attention toward relevant spans. They have experiments on multi-doc QA and a long-context benchmark HELMET, they show improved performance using various LLMs.

**Strengths:**

1. They address a well-known problem of distraction in long-context LLMs in a mechanistic perspective, which is interesting.
2. The proposed method adpots steering vector approach which is efficient.
3. The movtivation is clear by showing that contextual heads exist.

**Weaknesses:**

1. The idea of steering attention activateions has been well explored in several prior work, the proposed focus directions are another instance of attention steering, which lack novelty for ICLR conference.
2. The method heavily reply on gold spans or releveant documents in a dataset or domain to identify contextual heads and train focus directions. The transferability to different domain in zero shot  is limited.
3. Experiments only consifer synthetic distractors and a subset of HELMET. How the method could be generalized to multi-hop reasoning where there are multiple relevant spans?
4. Addressing distraction for long-context LLMs is a well-explored domain, however, baselines and related approaches are missing in this paper. For example, "Never Lost in the Middle" which directly address the distraction issue without training, and "Reducing distraction in long-context language models by focused learning" directly addresses distraction with training, which is more closely related.

**Questions:**

See weaknesses.

---

> ### Author Response · Authors · 2025-12-01
>
> W1: Attention steering lacks novelty
>
> Thanks for the comment. We do not consider our novelty as an “attention steering method”, but from the following aspects centered around focus directions:
>
> Contextual heads: We reveal that a particular group of attention heads controls the overall attention of LLMs to context.
>
> Attention activation control: We found that there exist directions (i.e., focus directions) at contextual heads that control the amount of attention activations from the attention sink to the contexts.
>
> Distraction mitigation mechanism: We found that proper control of attention activation by focus direction could make LLM pay more attention to the relevant contexts than the irrelevant ones, and thus mitigate the distractions. The effectiveness is strong in the domain and can also generalize to other domains to some extent.
>
>
> W2: relies on gold spans in domain, limited transferability
>
>
> Thanks for the comment. First, we clarify that our obtained focus directions have strong effectiveness in-domain (Section 3) and some transferability out-of-domain (Section 4). While the out-of-domain improvement is not as strong as in-domain, we consider this a basic property of focus directions. This work demonstrates the existence and effectiveness of focus direction under in-domain settings. Future work could further explore multiple focus directions for different domains and switching between different directions.
>
>
>
> W3: multi-hop/multiple relevant spans results
>
> Multi-hop/multiple relevant spans are actually covered in HELMET benchmarks (e.g., HotpotQA). Focus directions show reasonable improvement in such a case. For example, in Table 5, we show the result of Llama-3.2-3B improved from 52.67 (baseline) to 56.0.
>
>
> W4: lack of baselines of other methods
>
> We do not include baselines for two reasons:
>
> We consider focus directions as an internal mechanism of LLMs instead of a method to alleviate the distraction. The main goal of this paper is to study the properties of the focus directions. Developing methods to alleviate the distraction based on the focus directions could be explored in future work.
>
> Lack of standard benchmarks and runnable open source implementations. Despite a handful of works having been proposed, the field currently lacks standard benchmarks and runnable code for comparison.

---

### Official Review · Reviewer_rdFQ · 2025-11-01

**Soundness:** 3
**Presentation:** 3
**Contribution:** 3
**Rating:** 4
**Confidence:** 4

**Summary:**

The core contribution of this paper is a thorough systematic study that seeks to better understand why LLMs get distracted by irrelevant contexts in long-context settings. Through a controlled study, the authors demonstrate that by increasing the LLMs attention weights towards the relevant contexts can mitigate the propensity for LLMs to get distracted. Using this, they propose focus directions which allows the model to allocate more attention towards the relevant contexts, thereby improving performance.

**Strengths:**

- The study of the cause of LLMs distractions is novel and well-motivated, allowing for a better understanding of LLMs failures. I enjoyed reading section 2.
- The results from section 3 show clear improvements when using the proposed method, further justifying the validity of the method.

**Weaknesses:**

- While the model shows strong gains in section 3, the improvement in section 4 (on HELMET) are minimal, barely improving over the baseline. This suggests that the approach is limited to settings in which in-domain training data is available.
- The method to obtain focus directions depends on datasets where relevant and irrelevant contexts are annotated. This limits scalability and makes it hard to apply in real-world settings where such labels are unavailable.
- The performance highly depends on the magnitude parameter ($\alpha$). Too strong or too weak interventions can break the attention distribution and lead to performance drops. Although the authors propose automated magnitude control, it is still preliminary.
- Training and applying focus directions require caching key/query activations and modifying attention weights during inference, which can't be applied to modern techniques in inference speed up like FlashAttention2.

**Questions:**

- How would a simple two-step approach that first filters out relevant context with a relevance classifier and feeds that to an LLM compare?
- Can you provide some qualitative or visual analysis of how contextual heads or focus directions behave, and how they differ from other known functional heads like retrieval heads?

---

> ### Author Response · Authors · 2025-12-01
>
> W1: minimal improvement on HELMET, limited to in-domain training data.
>
> While we agree our obtained focus directions show some but limited improvement in out-of-domain tasks in HELMET, we consider this as a basic property of the focus directions. Similar to the no free lunch theorem, we shouldn’t expect a single direction to generalize well on every task. This work demonstrates the existence and effectiveness of focus direction under in-domain settings. Future work could further explore multiple focus directions for different domains and switching between different directions.
>
> W2: depends on annotated data, hard to apply in real-world settings
>
> While we agree that obtaining focus directions needs annotated data, we don’t think this could make it challenging to apply to real-world settings. Nowadays, every LLM fine-tuning work requires some sort of data construction under certain conditions. These data can be real or synthetic. Obtaining focus directions does not introduce additional data construction overhead compared to fine-tuning, and thus is still applicable to real-world settings under reasonable resource constraints.
>
>
> W3: performance depends on the magnitude parameter
>
> Thanks for pointing that out. While we agree that performance depends on the magnitude, we do not view this as a weakness. Instead, we consider this as a property of focus directions. Focus directions control the amount of attention activation from the attention sink to the contexts. Such amount of attention activation may not be optimal for some pre-trained or fine-tuned LLMs under certain tasks and thus result in not optimal downstream performance. In such cases, focus directions help to align LLMs to the optimal attention activation.
>
>
> W4: caching key/query activations cannot be applied to FlashAttention2
>
> There may be some misunderstanding here. Our focus directions are vectors to be added to the key/query activations before using any attention implementation, including FlashAttention2. Applying focus directions does not alter the attention implementation methods. Thus, focus directions are compatible with any attention implementation methods. In fact, our current code also runs on FlashAttention2.
>
>
> Q1: relevance classifier baseline
>
> Thanks for the comment. Determining relevant contexts is a challenging task. There is no off-the-shelf “relevance classifier” to be used for such purposes. To our understanding, the closest thing to the “relevance classifier” is the retrievers in RAG. Such retrievers usually only ensure high recall, but low precision. In our setting (Sections 2 and 3), the 20 documents are already obtained through retrievers.
>
> Q2: behavior of contextual heads/focus directions, different from retrieval heads
>
> Contextual heads: We visualize the contextual head in Figure 2 and the result of modifying attention on contextual heads in Figure 3.
> Focus directions: We show the attention distribution when applying focus directions in Table 3. We highlight that the focus directions could move the attention from the attention sink to the relevant contexts.
> Different from retrieval heads: We visualize the location of the contextual head vs the retrieval head in Figure 9.

---

### Note · Authors · 2026-01-06

I have read and agree with the venue's withdrawal policy on behalf of myself and my co-authors.